# FASA: Frequency-Aware Sparse Attention

**Yifei Wang**[1]     **Yueqi Wang**[2]     **Zhenrui Yue**[3]     **Huimin Zeng**[3]     **Yong Wang**[1*]
**Ismini Lourentzou**[3]     **Zhengzhong Tu**[4]     **Xiangxiang Chu**[1]     **Julian McAuley**[2]
[1] AMAP, Alibaba Group     [2] UCSD     [3] UIUC     [4] Texas A&M University

## Abstract

The deployment of Large Language Models (LLMs) faces a critical bottleneck when handling lengthy inputs: the prohibitive memory footprint of the Key Value (KV) cache. To address this bottleneck, the token pruning paradigm leverages attention sparsity to selectively retain a small, critical subset of tokens. However, existing approaches fall short, with static methods risking irreversible information loss and dynamic strategies employing heuristics that insufficiently capture the query-dependent nature of token importance. We propose FASA, a novel framework that achieves query-aware token eviction by dynamically predicting token importance. FASA stems from a novel insight into RoPE: the discovery of functional sparsity at the frequency-chunk (FC) level. Our key finding is that a small, identifiable subset of "dominant" FCs consistently exhibits high contextual agreement with the full attention head. This provides a robust and computationally free proxy for identifying salient tokens. Building on this insight, FASA first identifies a critical set of tokens using dominant FCs, and then performs focused attention computation solely on this pruned subset. Across a spectrum of long-context tasks, from sequence modeling to complex CoT reasoning, FASA consistently outperforms all token-eviction baselines and achieves near-oracle accuracy, demonstrating remarkable robustness even under constraint budgets. Notably, on LongBench-V1, FASA reaches nearly 100% of full-KV performance when only keeping 256 tokens, and achieves $2.56\times$ speedup using just 18.9% of the cache on AIME24 [1].

## 1 Introduction

Despite recent advances in Large Language Models (Dao et al., 2022; Ainslie et al., 2023; Liu et al., 2024a; Wang et al., 2025c) in long-context processing, requirements such as repository-level code analysis (Chen et al., 2021; Shi et al., 2025; Wang et al., 2026; Chen et al., 2025b) and document summarization (Goyal & Durrett, 2020) pose both memory and computational challenges, especially the linear growth of the KV cache. As the sequences grow, each token generation requires accessing the entire KV cache, leading to increased memory I/O latency. This memory-bound process underutilizes high-performance GPUs, ultimately limiting the overall throughput. To optimize KV cache management, previous studies have proposed mainly five directions: *token eviction* (Akhauri et al., 2025; Fang et al., 2025), *low-rank compression* (Chang et al., 2025; Singhania et al., 2024; Zhang et al., 2025), *quantization* (Hooper et al., 2025b; Liu et al., 2024d), *KV merging* (Wang et al., 2025d; Wan et al., 2025; Liu et al., 2024b), and *budget allocation* (Cai et al., 2025b).

An intuitive and widely explored approach is *token eviction* (LI et al., 2025; Liu et al., 2023). The rationale is that only a small subset of tokens contributes significantly to outputs, enabling the selective removal of trivial ones. Existing token eviction methods can be classified into three types: **(1)** *Static strategies* remove tokens with fixed rules (Xiao et al., 2024), therefore risking irreversible information loss; **(2)** *Adaptive strategies* either permanently evict less critical tokens (Zhang et al., 2023; Li et al., 2024) or preserve the full cache while retrieving a subset of entries (Tang et al., 2024; Ge et al., 2024). Yet such heuristic rankings provide an imperfect proxy for the truly dynamic nature of token importance; **(3)** *Learning-based strategies* (Akhauri et al., 2025; Yang et al., 2025; Chen et al., 2025a) rely on a trained token predictor, suffering from poor generalization on different datasets. *Can a token predictor achieve **query-awareness** without resorting to costly training?*

---

*Project lead and corresponding author.

[1] Code is available at https://github.com/AMAP-ML/FASA-ICLR2026

In response to this question, we introduce FASA (Frequency-Aware Sparse Attention), a **training-free**, **high-granularity**, **query-aware** predictor designed to evaluate token significance during the decoding phase, in a training-free manner. The design of FASA is rooted in an intriguing observation that differential frequencies within RoPE (Su et al., 2023) induce functional sparsity among frequency chunks (FCs). Only a sparse subset of FCs, termed as dominant FCs, contribute significantly to contextual awareness, while others construct robust positional patterns. We empirically verify that these dominant FCs are sparse, universal, and task-agnostic in Section 3.3, thereby providing a robust foundation for accurately predicting token importance.

Building upon this insight, FASA employs a two-stage framework for efficient inference. The first stage, Token Importance Prediction, harnesses dominant FCs to dynamically estimate attention scores, obtaining critical tokens. At the second stage, Focused Attention Computation then performs precise and focused token generation on this reduced set. The overhead of FASA is minimal because the identification of dominant FCs is a one-time and task–invariant process. Ultimately, FASA achieves high efficiency by fetching only a small fraction of the KV cache, which significantly reduces the data transferred between memory and the processor and thereby lowers memory bandwidth consumption. The overview of FASA is in Figure 2. Grounded on the same principles above, we introduce two variants of FASA: **FASA-M** and **FASA-C**. While they differ in implementation strategies, both *achieve equivalent downstream task performance* while offering different efficiency profiles, specializing in memory and computation, respectively. Crucially, despite FASA leverages a low-rank subspace, its primary objective is the dynamic prediction of token importance, not mere dimensionality reduction. This design makes FASA orthogonal to and compatible with most other KV cache compression methods. For example, it can be seamlessly integrated with layer-wise budget allocation schemes like PyramidKV (Cai et al., 2025b).

We evaluated FASA across a range of LLMs with varying KV cache budgets, concentrating on three core tasks: long-context benchmark, long-sequence modeling, and long chain-of-thought (LongCoT) reasoning. Our method achieves performance comparable to that of full KV cache, with reduction of less than 0.7%, while consistently surpassing all baseline methods across these tasks. FASA-M provides an $8\times$ compression of the KV cache, substantially optimizing memory usage. and FASA-C delivers $2.6\times$ speedups, enhancing computational efficiency, with 25% of FCs selected. Our contributions are summarized as follows:

- We are the first to uncover an intriguing finding: functional sparsity at FC-level induced by RoPE.
- Leveraging the functional sparsity of FCs, we introduce FASA, a training-free framework for dynamically predicting token importance.
- We present two variants of FASA: FASA-M, optimized for settings with memory constraints, and FASA-C, designed for scenarios with computational constraints.
- Extensive experiments across three paradigm tasks demonstrate that FASA consistently achieves near-oracle accuracy in both long-context and long-generation tasks.

## 2 RELATED WORKS

**Token Eviction.** A central theme in recent KV cache optimization (Hooper et al., 2025a; Wang et al., 2025a) is the exploitation of inherent, query-dependent attention sparsity (Liu et al., 2024c; 2025a; Behnam et al., 2025). Stream (Xiao et al., 2024) employs a rigid heuristic, preserving only initial and recent tokens, which invariably discards potentially crucial information from intermediate positions. SnapKV (Li et al., 2024) improves on this by introducing a one-time, prefill-stage filtering based on empirically estimated attention scores. However, the static nature of this estimation cannot adapt to the evolving relevance of tokens as generation progresses. Quest (Tang et al., 2024) offers a more dynamic solution by organizing the KV cache into pages and selectively fetching them. Despite its dynamism, its efficacy is hampered by a coarse, page-level granularity, which incurs significant overhead by forcing the retrieval of entire pages even when only a few tokens are needed.

**Low-rank Compression.** Another prominent paradigm for KV cache compression is low-rank approximation (Zhang et al., 2025; Dong et al., 2024), predicated on the observation that the cache's information content is concentrated in a low-dimensional subspace (Sun et al., 2025; sax, 2024; Behnam et al., 2025). For instance, SparQ (Ribar et al., 2024) employs a heuristic that selects key dimensions based on high query-vector magnitudes, a strategy that proves suboptimal due to its head-agnostic nature and its simplistic reliance on magnitude as a proxy for importance. Similarly,

LoKi (Singhania et al., 2024) leverages Principal Component Analysis (PCA) to project key states into a compact subspace for efficient computation, but at the cost of significant memory overhead from storing the requisite projection matrices. In contrast, our proposed FASA circumvents these limitations by operating in-place on the KV cache, thereby incurring no auxiliary memory overhead.

## 3 OBSERVATION

### 3.1 PRELIMINARY: ROTARY POSITIONAL ENCODINGS (ROPE)

RoPE embeds relative position information into the self-attention computation. Specifically, for a query vector $\mathbf{q}_{t_1}$ and a key vector $\mathbf{k}_{t_2}$ at positions $t_1$ and $t_2$, the attention score is formulated as $\mathbf{A}_{t_1, t_2} = (\mathbf{q}_{t_1} \mathbf{R}_{t_1})(\mathbf{k}_{t_2} \mathbf{R}_{t_2})^\top = \mathbf{q}_{t_1} \mathbf{R}_{\Delta t} \mathbf{k}_{t_2}^\top$. Due to the orthogonality, the product of $\mathbf{R}_{t_1}$ and $\mathbf{R}_{t_2}$ elegantly simplifies to a single rotation matrix parameterized solely by the relative offset $\Delta t = t_1 - t_2$.

**A Frequency-Chunk Perspective on RoPE.** From a frequency-domain perspective, the RoPE mechanism can be interpreted through the concept of "frequency chunks" (FCs). This framework posits that any $d$-dimensional vector $\mathbf{v} \in \mathbb{R}^d$ (e.g., a query and key) is partitioned into $d/2$ orthogonal 2D subspaces. We denote the $i$-th such subspace, or FC, as $\mathbf{v}^{[i]} = (v_{2i}, v_{2i+1})^T$. Each FC is associated with a unique base angular frequency, calculated as $\theta_i = B^{-2(i-1)/d}$ for $i \in \{1, \dots, d/2\}$, where $B$ is a predefined frequency base. This design establishes a direct mapping from a chunk's dimensional indices $(2i, 2i+1)$ to its rotational frequency. *Lower dimension indices (i) result in higher frequencies, which implies that the corresponding FCs rotate very quickly physically.* For a token at absolute position $m$, its $i$-th FC is rotated by an angle $m\theta_i$ through a specific $2 \times 2$ rotation matrix $\mathbf{R}_{m,\theta_i}$. The global rotation matrix $\mathbf{R}_{\Delta t}$ is block-diagonal, where each diagonal block is a $2 \times 2$ rotation matrix $\mathbf{R}_{\Delta t, \theta_i}$ and defined as $\mathbf{R}_{\Delta t} = \mathrm{Diag}(\mathbf{R}_{\Delta t, \theta_1}, \mathbf{R}_{\Delta t, \theta_2}, \dots, \mathbf{R}_{\Delta t, \theta_{d/2}}) = \bigoplus_{i=1}^{d/2} \mathbf{R}_{\Delta t, \theta_i}$.

$$\mathbf{v}_m = \bigoplus_{k=1}^{d/2} \mathbf{v}_m^{[i]} = \bigoplus_{k=1}^{d/2} (\mathbf{v}_{2i}, \mathbf{v}_{2i+1})^T, \mathbf{R}_{m,\theta_i} = \begin{pmatrix} \cos(m\theta_i) & -\sin(m\theta_i) \\ \sin(m\theta_i) & \cos(m\theta_i) \end{pmatrix}. \quad (1)$$

### 3.2 MOTIVATION AND HYPOTHESIS

**Position vs. Semantics: Different Roles of FCs.** The varying rotational velocities across FCs inherently lead to functional heterogeneity. This principle is substantiated by two key observations from prior literature. First, a distinct division of labor exists within RoPE (Barbero et al., 2025; Wei et al., 2025), where high-frequency FCs (in low dimensions) are primarily responsible for constructing robust positional patterns, and in contrast, low-frequency counterparts specialize in carrying the semantic information and model long-range dependencies. Second, this functional specialization is structurally reflected by a RoPE-induced concentration of high-magnitude values within specific query and key dimensions (Sun et al., 2024), reinforcing the non-uniform functional importance of FCs. This functional heterogeneity suggests that FCs can be grouped into two distinct categories:

1. **Contextual FCs:** A small, critical subset responsible for dynamic, context-specific attention. These FCs identify which tokens are semantically relevant to the current query.
2. **Structural FCs:** The remaining majority primarily injects inherent, positional attention patterns, mainly recency bias (Peysakhovich & Lerer, 2023) and attention sinks (Xiao et al., 2024).

**Hypothesis:** *The model's contextual awareness is overwhelmingly driven by the Contextual FCs. A few contextual FCs could replicate the contextual selection behavior of a full attention head.* If their index set is denoted as $\mathcal{I}_{\mathrm{dom}} \subset \{1, \dots, d/2\}$, the full attention dot product can be effectively approximated by summing only over $\mathcal{I}_{\mathrm{dom}}$, namely $\mathbf{A}_{t_1, t_2} = \mathbf{q}_{t_1} \mathbf{R}_{\Delta t} \mathbf{k}_{t_2}^T \sum_{i \in \mathcal{I}_{\mathrm{dom}}} \mathbf{q}_{t_1}^{[i]} \mathbf{R}_{\Delta t, \theta_i} \mathbf{k}_{t_2}^{[i]\top}$.

### 3.3 QUANTIFYING FUNCTIONAL SPARSITY

Quantifying our hypothesis of FC-level functional sparsity requires a metric to assess the "dominance" of individual FCs. Therefore, we propose the **Contextual Agreement (CA)** metric, which measures the alignment between the attention pattern from a single FC and that of the full attention head.

**Formal Setup.** For a query $\mathbf{q}_t \in \mathbb{R}^d$ and key matrix $\mathbf{K}_{1:t} \in \mathbb{R}^{d \times t}$ in an attention head $(l, h)$, we define two raw score vectors: the standard **full-head scores** $\boldsymbol{\alpha}_{l,h}$ and the **single-FC scores** $\boldsymbol{\alpha}_{l,h}^{(i)}$. The

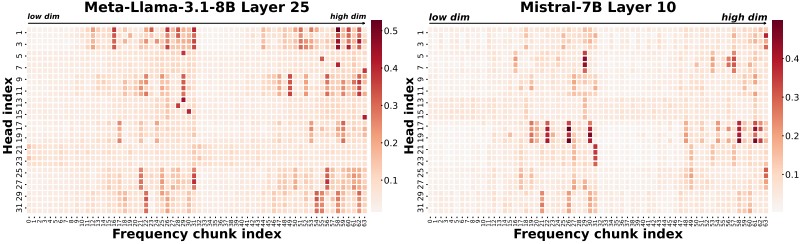

Figure 1: **Functional sparsity of FCs revealed by Contextual Agreement ($\overline{\text{CA}}$) heatmaps.** Each heatmap shows $\overline{\text{CA}}$ per FC ($x$-axis) across all heads ($y$-axis). A few "dominant" FCs (bright vertical bands) consistently capture contextual information across attention heads. Results on Qasper ($\mathcal{K} = 256$); see Appendix A.

latter are computed using only the 2D components of the $i$-th FC. These are expressed as:

$$\boldsymbol{\alpha}_{l,h}(\mathbf{q}_t, \mathbf{K}_{1:t}) = [\mathbf{q}_t \, \mathbf{R}_{t-1} \, (\mathbf{k}_0)^T, \cdots, \mathbf{q}_t \, \mathbf{R}_0 \, (\mathbf{k}_t)^T]^T \tag{2}$$

$$\boldsymbol{\alpha}_{l,h}^{(i)}(\mathbf{q}_t, \mathbf{K}_{1:t}) = [\mathbf{q}_t^{[i]} \, \mathbf{R}_{t-1,\theta_i} \, \mathbf{k}_0^{[i]^T}, \cdots, \mathbf{q}_t^{[i]} \, \mathbf{R}_{0,\theta_i} \, \mathbf{k}_t^{[i]^T}]^T \tag{3}$$

**Metric Definition.** The **CA** score, $\text{CA}_{\mathcal{K}}^{l,h,i}$, quantifies the agreement between the full-head $\boldsymbol{\alpha}_{l,h}$ and single-FC $\boldsymbol{\alpha}_{l,h}^{(i)}$ scores by measuring the normalized intersection of their top-$\mathcal{K}$ token index sets:

$$\text{CA}_{\mathcal{K}}^{l,h,i}(q_t, \mathbf{K}_{1:t}) = [\text{TopK-I}(\boldsymbol{\alpha}_{l,h}(q_t, \mathbf{K}_{1:t}), \mathcal{K}) \cap \text{TopK-I}(\boldsymbol{\alpha}_{l,h}^{(i)}(q_t, \mathbf{K}_{1:t}), \mathcal{K})]/\mathcal{K}, \tag{4}$$

where the operator $\text{TopK-I}(\boldsymbol{\alpha}, \mathcal{K})$ retrieves the top-$\mathcal{K}$ values of a vector $\boldsymbol{\alpha}$. To assess an FC's importance robustly, we compute its mean CA score, by averaging across several samples from a specific dataset. Figure 1 reveals the distinct functional contribution of each FC across all heads.

**Sparse and Universal $\mathcal{I}_{\text{dom}}$.** **(1)** *Sparsity*: A small subset of FCs (dominant FCs) exhibits disproportionately high agreement with full attention patterns, while the vast majority of other FCs have negligible CA scores (typically $< 0.15$). In Table 9, dominant FCs account for less than 1% of all FCs, while non-dominant FCs with low CA scores comprise approximately 90% or more; **(2)** *Universality*: The functional sparsity is widely observed across model architectures and scales (Appendix A.1;Table 9); **(3)** *Task-Invariance:* The identification of dominant FCs is largely task-agnostic. The

Table 1: Compound CA scores under varying number of selected FCs ($F$) and KV cache budgets ($K$). Each head has 64 FCs in total.

| $\dfrac{K}{\|\mathcal{I}_{dom}\|}$ | 64 | 256 | 512 | 768 | 1024 | 2048 |
|---|---|---|---|---|---|---|
| Random | 2.0 | 3.6 | 6.4 | 19.1 | 25.5 | 51.1 |
| Stream | 34.4 | 26.8 | 24.4 | 26.5 | 30.7 | 53.9 |
| SnapKV | 37.9 | 40.9 | 41.9 | 45.4 | 49.5 | 66.6 |
| $F = 8$ (1/8) | 43.0 | 49.4 | 54.3 | 58.8 | 62.6 | 76.1 |
| $F = 10$ | 46.4 | 52.1 | 56.6 | 61.1 | 64.8 | 77.5 |
| $F = 12$ | 49.7 | 54.7 | 58.9 | 63.4 | 66.8 | 79.0 |
| $F = 14$ | 52.4 | 56.9 | 60.9 | 65.2 | 68.5 | 80.2 |
| $F = 16$ (1/4) | 55.3 | 59.7 | 62.8 | 66.9 | 70.1 | 81.4 |

saliency maps in Figure 12 derived from tasks such as QA and summarization exhibit remarkable consistency. Quantitatively, the overlap of dominant FCs across different calibration datasets consistently exceeds 70% in all tested models, as reported in Table 10. This indicates that the functional roles of FCs are intrinsic to the underlying mechanics of RoPE, rather than being task-specific adaptations.

**Reconstructing Functionality from $\mathcal{I}_{\text{dom}}$.** The analysis above supports that the functionality of a full attention head can be reconstructed using only its most dominant $F$ components $\mathcal{I}_{\text{dom}}^{l,h} = \text{TopK-I}(\{\text{CA}_{\mathcal{K}}^{l,h,i} \mid 0 \leq f < d/2\}, F)$. Therefore, we measure the collective efficacy of this subset using a compound CA score, $\text{CA}_{K}^{l,h,\mathcal{I}_{\text{dom}}}$, and present the results in Table 1. For comparison, we benchmark against token-eviction methods, which serve to emphasize the capability of predicting token importance. Our method demonstrates remarkable efficiency: with just 1/8 of the components selected under a tight budget 64, $\mathcal{I}_{\text{dom}}$ achieves an accuracy of 43%, surpassing the strong baseline SnapKV (Li et al., 2024) by an average of 10.3% across all budget levels. We also present the predictive distribution of dominant FCs across tokens grouped by attention score quintiles in Table 11. This analysis further substantiates the ability of dominant FCs to effectively capture both the relative ranking and true impact of context tokens.

## 4 METHOD

Grounded in the functional sparsity of FCs, our training-free framework FASA employs a two-stage, coarse-to-fine strategy to circumvent the prohibitive cost of full self-attention. First, the

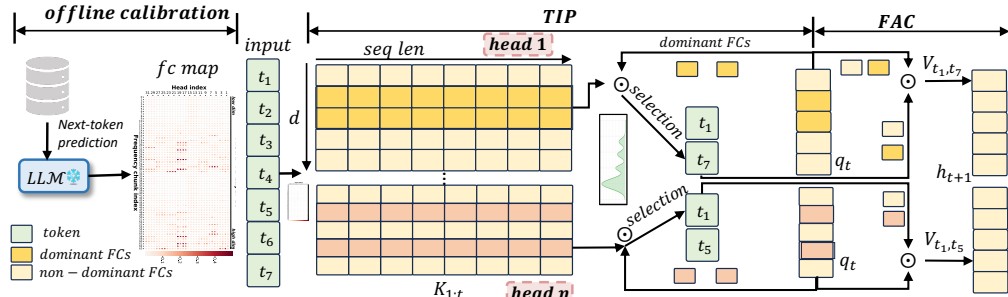

Figure 2: Method Overview of FASA. First, the **TIP** stage leverages only dominant FCs to efficiently estimate token importance and select a critical subset of tokens. Then, the **FAC** stage performs full-dimensional attention exclusively on this reduced subset to generate the next token. See discussion about design in Appendix D.2.

**Token Importance Predictor (TIP)** stage utilizes a computationally frugal proxy, defined by a pre-calibrated set of dominant FCs, $\mathcal{I}_{\text{dom}}$, to efficiently identify a small subset of contextually salient tokens. Subsequently, the **Focused Attention Computation (FAC)** stage performs a full-fidelity attention computation exclusively on this salient subset, preserving high generation fidelity while drastically mitigating the computational and memory overhead of standard attention.

## 4.1 Token Importance Predictor (TIP)

The TIP stage operates on the principle that dominant frequencies are an efficient proxy for token importance, where the dominant indices $\mathcal{I}_{dom}$ are identified via a one-time offline calibration.

**Offline Calibration: Identifying $\mathcal{I}_{dom}$.** The objective of the offline calibration is to identify a small, head-specific set of *dominant frequencies*, $\mathcal{I}_{\text{dom}}^{l,h}$, for each attention head $(l, h)$. We formulate this process as a search problem over frequency indices. Given a small calibration dataset $\Omega$ and a target size $N_{tip}$, our goal is to find the subset of FCs of cardinality $N_{tip}$ that maximizes the expected average of CA scores. The objective is defined as:

$$\mathcal{I}_{\text{dom}}^{l,h} = \underset{\mathcal{I} \subseteq \{0,\ldots,d/2-1\}, |\mathcal{I}|=N_{tip}}{\operatorname{argmax}} \mathbb{E}_{\mathbf{q},\mathbf{K} \sim \Omega} \left[ \sum_{i \in \mathcal{I}} \text{CA}_{\mathcal{K}}^{l,h,i}(\mathbf{q}, \mathbf{K}) \right]. \tag{5}$$

This calibration is a highly efficient, one-time offline process because the resulting $\mathcal{I}_{\text{dom}}$ is empirically found to be task-agnostic and can be robustly identified from a minimal number of samples. Its associated computational cost is negligible. The detailed algorithm is provided in Algorithm 1.

**Online Prediction: Importance Scoring via Frequency Subspace Aggregation.** During the online prediction phase at a given decoding step $t$, we leverage the pre-calibrated set of dominant frequencies, $\mathcal{I}_{\text{dom}}^{l,h}$, to efficiently estimate token importance in a training-free manner. Conceptually, the full attention score for a query $\mathbf{q}_t$ and keys $\mathbf{K}_{1:t}$ can be decomposed into a sum of contributions from all $d/2$ frequency components: $\boldsymbol{\alpha}^{l,h}(\mathbf{q}_t, \mathbf{K}_{1:t}) = \sum_{i=0}^{d/2-1} \boldsymbol{\alpha}^{l,h,i}(\mathbf{q}_t, \mathbf{K}_{1:t})$. Instead of performing this computationally expensive summation, our method constructs an *importance score vector* $\mathbf{S}_t^{l,h}$, by exclusively aggregating the contributions from the pre-identified dominant frequencies, i.e., $\mathbf{S}_t^{l,h} \triangleq \sum_{i \in \mathcal{I}_{\text{dom}}^{l,h}} \boldsymbol{\alpha}^{l,h,i}(\mathbf{q}_t, \mathbf{K}_{1:t})$. This formulation strategically bypasses computation for non-dominant frequencies. Finally, based on these scores, we identify the set of top-$N_{fac}$ most important token indices, $\mathcal{T}_t$, for the subsequent FAC stage: $\mathcal{T}_t = \text{TopK-I}(\mathbf{S}_t^{l,h}, N_{fac})$.

## 4.2 Focused Attention Computation (FAC)

Following the identification of the contextually important token set $\mathcal{T}_t$ by the TIP module, this stage executes an attention computation on $\mathcal{T}_t$, enabling the model to concentrate its computational resources on the most salient parts of the context. Specifically, for the current query vector $\mathbf{q}_t$ at decoding step $t$, instead of using the full key and value matrices $(\mathbf{K}_{1:t}, \mathbf{V}_{1:t})$ from the entire past context, we first gather the keys and values corresponding to the indices in $\mathcal{T}_t$:

$$\mathbf{K}_{\mathcal{T}_t} = \text{Gather}(\mathbf{K}_{1:t}, \mathcal{T}_t), \quad \mathbf{V}_{\mathcal{T}_t} = \text{Gather}(\mathbf{V}_{1:t}, \mathcal{T}_t) \tag{6}$$

where the Gather$(\cdot)$ operation selects the rows from the original matrices specified by the index set $\mathcal{T}_t$. The attention scores for each head $(l, h)$ are then computed using only these selected keys. The final output vector for the head is subsequently produced by weighting the selected value vectors:

$$\hat{\boldsymbol{\alpha}}^{l,h}_{\text{FAC}} = \text{Softmax}\left(\mathbf{q}_t \mathbf{K}_{\mathcal{T}_t}{}^T / \sqrt{d}\right), \quad \mathbf{O}^{l,h}_t = \hat{\boldsymbol{\alpha}}^{l,h}_{\text{FAC}} \mathbf{V}_{\mathcal{T}_t} \tag{7}$$

Critically, the original absolute positions of the tokens in $\mathcal{T}_t$ are preserved. This directly maintains the integrity of their position embeddings and the vital spatial information they encode, preventing the performance degradation associated with positional distortion. In essence, the FAC stage functions as a high-fidelity computational filter, restricting full-precision attention to the most salient tokens to achieve a compelling balance between computational efficiency and predictive accuracy.

### 4.3 Two Implementations of FASA

We introduce two specialized, hardware-aware variants of FASA that offer a trade-off between memory and speed: (1). **FASA-M (Memory-Optimized)** minimizes its GPU memory footprint by strategically offloading the value cache and non-dominant key components to CPU memory, making it ideal for VRAM-constrained environments. To mitigate the latency from CPU-GPU data transfer, this approach can be effectively paired with prefetching techniques. (2) **FASA-C (Computation-Optimized)** prioritizes inference speed by retaining the full cache on-GPU but accessing only a sparse subset of key states, drastically reducing memory I/O for significant acceleration. (See Appendix D.1 for details and memory analysis of FASA-M).

### 4.4 Efficiency Analysis of FASA

**Computational Analysis.** At the generation step $t$, the complexity of computing $\mathbf{q}_t \mathbf{K}^{\mathbf{T}}_{1:t}$ is $\mathcal{O}(td)$ and the complexity of multiplying the value states with attention scores is $\mathcal{O}(td)$ per head. For FASA, (1) the complexity of the **TIP** stage is $\mathcal{O}(2tN_{\text{tip}})$ (each FC takes up 2 dimensions), since this stage operates in low-dimensional subspaces, and (2) the **FAC** stage performs attention on a reduced set of $N_{fac}$ tokens, leading to a complexity of $\mathcal{O}(N_{fac}d)$. Additionally, the detection of dominant frequencies $\mathcal{I}_{dom}$ is offline, one-time, and applicable for various tasks and the burdens from this part could be neglected. Assuming the complexity of selecting the top-k tokens is small, the overall complexity of FASA is $\mathcal{O}(2tN_{tip} + 2N_{fac}d)$. The theoretical speedup at decoding stage is in Equation 8.

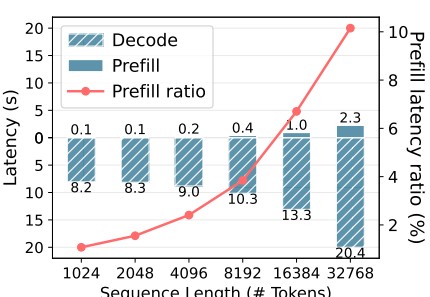

Figure 3: Decoding latency dominates total latency in auto-regressive generation.

$$\text{Speedup} = \frac{2td}{2tN_{tip} + 2N_{fac}d} = \frac{1}{N_{tip}/d + N_{fac}/t}, \text{Speedup} \approx \frac{d}{N_{tip}} \; if \; N_{fac} \ll t \tag{8}$$

**Memory Movement Reduction.** The auto-regressive decoding stage is notoriously memory-bound, as requiring loading the entire KV cache, creating a significant latency bottleneck. This is confirmed in Figure 3, where decoding constitutes $90\%$ of the total latency at a 32K context. FASA, directly mitigates this bottleneck by drastically reducing memory traffic. At a decoding step $t$, standard attention loads $2tm$ bytes from the KV cache (with $m$ as the byte size per state vector) while FASA accesses only $t(2N_{tip}/d * m)$ bytes (only keys) for the TIP and $2N_{\text{fac}}m$ bytes for the FAC. The fraction that FASA must load is therefore: $(2tmN_{tip}/d + 2N_{fac}m)/2tm = N_{tip}/d + N_{fac}/t \approx N_{tip}/d(N_{fac} \ll t)$, which alleviates the memory-bound constraint of long-context decoding.

## 5 Experiments

### 5.1 Experimental Setting

**Baselines and Models.** To comprehensively evaluate FASA's performance, we benchmark it against into two groups of robust baselines: **(1) State-of-the-art methods:** We compare against leading token eviction methods in efficient KV cache management, including Stream (Xiao et al., 2024),

Table 2: Performance of FASA on diverse models on LongBench-V1 benchmarks. For baselines, we retain constant token budget (256) and 25% FCs for FASA. †FKV and Oracle are full and look-ahead upper bounds.

| | Method | Single-Doc QA | | | Multi-Doc QA | | | Summarize | | | Summarize | | Synthetic | | Code | | AVG. |
|---|---|---|---|---|---|---|---|---|---|---|---|---|---|---|---|---|---|
| | | NQA | Qasp | MF-en | Hqa | 2Wiki | Musi | GovR | Qsum | Mult | Trec | Tqa | Pcnt | Pre | Lcc | RB-P | |
| **Llama3.2-3B** | FKV† | 26.0 | 40.7 | 50.4 | 32.2 | 29.6 | 15.1 | 33.5 | 22.9 | 25.3 | 71.5 | 88.9 | 3.5 | 87.8 | 52.0 | 54.2 | 42.2 |
| | Oracle† | 26.6 | 41.2 | 49.8 | 31.9 | 29.9 | 16.2 | 32.6 | 22.2 | 25.0 | 71.5 | 89.3 | 3.5 | 88.0 | 53.7 | 54.4 | 42.4 ↑0.2 |
| | Quest | 8.7 | 19.5 | 23.6 | 12.9 | 15.9 | 6.5 | 23.3 | 18.1 | 25.1 | 34.5 | 52.9 | 6.5 | 38.3 | 53.7 | 43.6 | 25.5 ↓16.7 |
| | Stream | 13.2 | 19.7 | 23.6 | 18.1 | 22.7 | 7.8 | 18.2 | 17.9 | 17.9 | 49.0 | 83.7 | 3.5 | 85.7 | 49.3 | 45.9 | 31.8 ↓10.4 |
| | SnapKV | 23.5 | 28.9 | 45.6 | 17.7 | 22.9 | 11.8 | 21.7 | 20.9 | 21.1 | 61.0 | 88.5 | 3.5 | 88.0 | 50.7 | 48.6 | 37.0 ↓5.2 |
| | FASA | 25.6 | 38.9 | 49.9 | 29.7 | 31.2 | 14.8 | 28.0 | 24.2 | 26.1 | 71.5 | 89.2 | 3.6 | 86.9 | 53.2 | 50.5 | 41.5 ↓0.7 |
| **Qwen2.5-7B** | FKV | 24.2 | 43.5 | 52.1 | 55.9 | 46.9 | 28.6 | 31.8 | 23.1 | 23.9 | 71.5 | 89.3 | 7.5 | 92.0 | 60.2 | 66.5 | 47.8 |
| | Oracle | 24.4 | 43.0 | 52.3 | 57.8 | 46.9 | 30.1 | 31.6 | 23.9 | 24.1 | 72.5 | 89.7 | 8.0 | 100.0 | 60.5 | 65.3 | 48.7 ↑0.9 |
| | Quest | 9.1 | 24.5 | 30.4 | 24.7 | 24.1 | 8.8 | 26.8 | 19.9 | 24.4 | 41.8 | 66.7 | 4.4 | 77.6 | 46.5 | 42.0 | 31.4 ↓16.4 |
| | Stream | 18.1 | 24.2 | 26.5 | 41.2 | 36.4 | 17.3 | 18.4 | 18.3 | 15.4 | 45.0 | 82.9 | 8.5 | 24.0 | 49.6 | 52.2 | 31.9 ↓15.9 |
| | SnapKV | 26.6 | 36.0 | 50.8 | 55.6 | 43.8 | 26.5 | 21.9 | 21.9 | 19.3 | 58.0 | 86.2 | 8.0 | 98.5 | 55.6 | 60.6 | 42.6 ↓5.2 |
| | FASA | 28.3 | 43.8 | 51.9 | 57.4 | 46.0 | 30.1 | 31.2 | 22.8 | 24.3 | 72.0 | 89.4 | 8.0 | 99.5 | 60.3 | 64.0 | 47.9 ↑0.1 |
| **stral-7B-v0.3** | FKV† | 29.1 | 41.6 | 52.9 | 49.4 | 39.5 | 29.1 | 34.8 | 25.7 | 27.8 | 76.0 | 88.6 | 5.5 | 98.0 | 58.4 | 59.7 | 47.4 |
| | Oracle† | 31.0 | 40.2 | 52.4 | 50.3 | 39.4 | 28.8 | 34.0 | 25.74 | 27.2 | 76.0 | 89.4 | 5.0 | 98.0 | 59.3 | 61.0 | 47.9 ↑0.5 |
| | Quest | 15.7 | 30.7 | 41.0 | 37.4 | 27.1 | 11.9 | 29.3 | 21.3 | 26.6 | 57.0 | 80.7 | 5.0 | 85.5 | 56.9 | 53.0 | 38.6 ↓8.8 |
| | Stream | 11.8 | 15.3 | 20.9 | 32.1 | 27.1 | 10.6 | 20.2 | 17.3 | 20.1 | 44.5 | 69.0 | 1.6 | 3.2 | 56.5 | 49.8 | 26.7 ↓20.7 |
| | SnapKV | 25.5 | 32.6 | 53.7 | 48.4 | 37.3 | 25.9 | 22.7 | 23.6 | 23.1 | 62.5 | 89.4 | 6.5 | 94.5 | 57.3 | 57.0 | 44.0 ↓3.4 |
| | FASA | 29.9 | 42.3 | 53.7 | 51.1 | 39.1 | 28.7 | 34.0 | 24.8 | 28.2 | 76.0 | 89.4 | 5.0 | 98.0 | 57.8 | 58.0 | 47.8 ↑0.4 |
| **Llama3.1-8B** | FKV† | 30.0 | 45.3 | 55.6 | 55.8 | 43.7 | 30.2 | 35.1 | 25.4 | 27.0 | 72.5 | 91.7 | 7.1 | 99.5 | 63.0 | 56.3 | 48.7 |
| | Oracle† | 30.3 | 44.5 | 55.0 | 54.9 | 44.6 | 32.0 | 34.8 | 25.1 | 26.9 | 72.5 | 91.5 | 7.0 | 99.5 | 63.3 | 57.4 | 48.7 ↓0.0 |
| | Quest | 13.7 | 33.1 | 38.4 | 35.8 | 32.2 | 12.8 | 26.5 | 20.9 | 26.7 | 38.0 | 65.6 | 3.8 | 95.0 | 52.5 | 45.7 | 35.4 ↓13.3 |
| | Stream | 21.9 | 23.4 | 31.8 | 45.1 | 36.7 | 24.3 | 20.0 | 21.0 | 19.3 | 45.5 | 87.9 | 6.9 | 99.5 | 59.4 | 49.1 | 38.8 ↓9.9 |
| | SnapKV | 27.5 | 34.5 | 51.6 | 52.3 | 44.3 | 28.3 | 23.9 | 24.0 | 22.7 | 62.5 | 90.9 | 7.5 | 99.5 | 60.1 | 52.6 | 45.0 ↓3.7 |
| | FASA | 29.3 | 43.7 | 54.1 | 54.8 | 43.9 | 30.8 | 33.5 | 24.7 | 27.0 | 72.0 | 91.1 | 7.5 | 99.5 | 61.8 | 52.7 | 48.2 ↓0.5 |
| **en2.5-14B-1M** | FKV† | 28.7 | 46.2 | 53.8 | 65.2 | 64.5 | 43.6 | 43.5 | 23.3 | 22.7 | 80.5 | 89.5 | 11.0 | 100.0 | 32.3 | 37.5 | 50.3 |
| | Oracle† | 28.5 | 46.3 | 54.3 | 64.3 | 63.6 | 44.7 | 31.5 | 22.9 | 22.7 | 81.0 | 88.4 | 10.0 | 100.0 | 33.6 | 39.7 | 49.4 ↓0.9 |
| | Quest | 14.5 | 31.9 | 39.1 | 38.8 | 36.6 | 16.2 | 16.2 | 20.1 | 25.2 | 43.5 | 72.7 | 10.0 | 88.8 | 35.0 | 34.0 | 34.9 ↓15.4 |
| | Stream | 19.6 | 26.9 | 29.4 | 46.5 | 48.3 | 29.6 | 17.8 | 18.4 | 15.0 | 46.5 | 82.5 | 12.5 | 72.1 | 28.7 | 31.2 | 35.3 ↓15.0 |
| | SnapKV | 26.3 | 40.5 | 51.2 | 63.2 | 62.2 | 43.3 | 22.5 | 22.0 | 18.3 | 63.5 | 87.5 | 11.5 | 100.0 | 30.4 | 36.0 | 45.9 ↓4.4 |
| | FASA | 27.2 | 45.5 | 54.5 | 64.4 | 63.9 | 44.5 | 30.4 | 22.8 | 21.9 | 80.0 | 87.5 | 15.5 | 100.0 | 30.5 | 36.1 | 49.2 ↓1.1 |

SnapKV (Li et al., 2024), RKV (Cai et al., 2025a), Quest (Tang et al., 2024), H2O (Zhang et al., 2023); **(2) Upper bounds:** two theoretical bounds, FKV, which represents standard inference with the complete, uncompressed KV cache, serving as the absolute performance ceiling due to no information loss, and Oracle, a more pragmatic upper bound for eviction-based methods, assuming ideal knowledge to retain only the most critical tokens based on full-head scores. Our experiments span a variety of cutting-edge architectures and model sizes, specifically Llama (Touvron et al., 2023), Mistral (Jiang et al., 2023), and Qwen (Bai et al., 2023).

**Evaluation Benchmarks.** To rigorously assess the capabilities of FASA across diverse long-context scenarios (Liu et al., 2025b), we conduct comprehensive evaluations spanning three paradigms: (1) **Long-context understanding:** We use diverse, real-world tasks from LongBench (Bai et al., 2024) to assess the ability to identify critical information within lengthy contexts. (2) **Long-Sequence Modeling:** We measure perplexity on PG-19 (Rae et al., 2019), WikiText (Merity et al., 2017), and C4 (Raffel et al., 2019) corpus to evaluate generative fidelity over long dependencies. (3) **Long-CoT Reasoning:** To test performance in long-generation scenarios, we evaluate on complex mathematical reasoning tasks from MATH500 (Hendrycks et al., 2021) and AIME24 (MAA, 2024) on R1-LLMs.

## 5.2 PERFORMANCE COMPARISON ON LONG-CONTEXT TASKS.

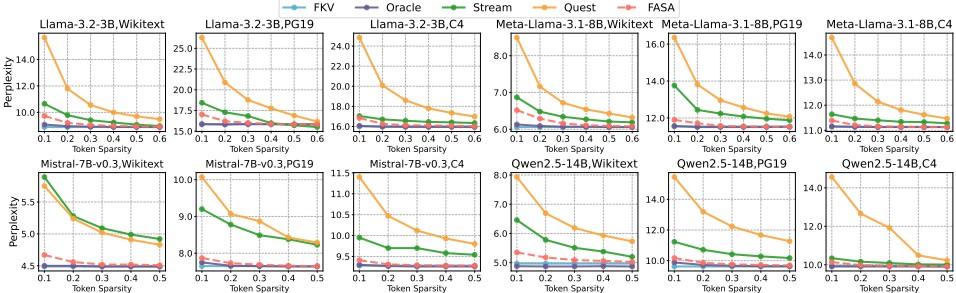

Figure 4: Perplexity results of FASA in comparison with FKV, Oracle, Stream, and Quest on Wikitext (**top**), PG19 (**middle**), and C4 corpus (**bottom**). Token sparsity indicates the retained ratio of tokens.

Table 3: Performance and output length of FASA compared to baseline models on the MATH500 and AIME24 $N_{tip} = 16$. AIME24 results are reported as pass@1, based on 16 responses per question. PREF* and DEC* denote the prefill and decoding lengths, respectively. †FKV and Oracle are full and look-ahead upper bounds.

| Methods | MATH500 | | | | | | | AIME24 | | | | | | | |
|---|---|---|---|---|---|---|---|---|---|---|---|---|---|---|---|
| | Fixed Budget | | | | Len Stats | | | Fixed Budget | | | | | Len Stats | | |
| | 300 | 500 | 700 | 1000 | PREF* | DEC* | TOTAL. | 500 | 1000 | 1500 | 2000 | 2500 | PREF* | DEC* | TOTAL. |
| DeepSeek-R1-Distill-Llama-8B | | | | | | | | | | | | | | | |
| FKV† | 72.4 | - | - | 72.4 | | 2977 | 3104 | 43.9 | - | - | - | 43.9 | | 13231 | 13392 |
| Oracle† | 70.4 | 72.6 | 74.2 | 71.8 | 127 | 3195 | 3321 | 30.0 | 36.7 | 37.3 | 39.3 | 36.0 | 161 | 15638 | 15799 |
| H2O | 6.8 | 33.0 | 53.87 | 42.8 | | 8244 | 8370 | 0.7 | 4.7 | 11.3 | 14.0 | 20.0 | | 21099 | 21260 |
| Stream | 9.6 | 24.6 | 40.4 | 47.4 | | 3520 | 3647 | 0.0 | 3.3 | 8.0 | 10.7 | 15.3 | | 10191 | 10352 |
| SnapKV | 21.6 | 32.6 | 46.8 | 54.6 | | 7047 | 7174 | 4.0 | 8.0 | 16.0 | 23.3 | 29.1 | | 17359 | 17520 |
| RKV | 24.0 | 39.4 | 49.2 | 57.0 | | 7005 | 7132 | 6.7 | 10.7 | 14.0 | 21.7 | 23.3 | | 22916 | 23077 |
| FASA | 62.2 | 68.8 | 69.4 | 71.8 | | 3171 | 3298 | 20.6 | 34.4 | 40.2 | 35.8 | 38.0 | | 17166 | 17327 |
| DeepSeek-R1-Distill-Qwen-14B | | | | | | | | | | | | | | | |
| FKV† | 92.4 | - | - | 92.4 | | 2784 | 2914 | 66.6 | - | - | - | 66.6 | | 11039 | 11204 |
| Oracle† | 92.2 | 92.4 | 92.4 | 92.2 | 127 | 2985 | 3112 | 67.9 | 66.7 | 67.3 | 70.7 | 67.3 | 165 | 11546 | 11711 |
| H2O | 29.6 | 50.2 | 62.8 | 77.0 | | 3413 | 3540 | 5.3 | 20.5 | 37.3 | 46.0 | 52.7 | | 9519 | 9684 |
| Stream | 27.8 | 44.0 | 57.8 | 64.4 | | 2801 | 2928 | 2.0 | 4.0 | 16.7 | 22.7 | 29.3 | | 8468 | 8633 |
| SnapKV | 34.2 | 55.8 | 69.4 | 79.4 | | 3586 | 3713 | 10.0 | 23.3 | 40.0 | 46.0 | 52.7 | | 11922 | 12083 |
| RKV | 57.8 | 74.0 | 80.8 | 86.4 | | 3865 | 3992 | 20.7 | 30.0 | 46.7 | 55.4 | 62.0 | | 16274 | 16439 |
| FASA | 86.6 | 88.8 | 90.2 | 91.2 | | 3139 | 3266 | 54.0 | 60.6 | 59.3 | 62.7 | 63.3 | | 11553 | 11709 |
| DeepSeek-R1-Distill-Qwen-32B | | | | | | | | | | | | | | | |
| FKV† | 92.6 | - | - | 92.6 | | 2717 | 2846 | 72.8 | - | - | - | 72.8 | | 10461 | 10626 |
| Oracle† | 92.4 | 91.4 | 91.4 | 91.2 | 127 | 2886 | 3013 | 68.0 | 70.1 | 70.0 | 76.7 | 69.2 | 156 | 11545 | 11710 |
| H2O | 47.2 | 50.0 | 62.8 | 74.4 | | 3841 | 3968 | 6.7 | 16.7 | 38.4 | 46.0 | 55.6 | | 10904 | 11069 |
| Stream | 43.6 | 57.6 | 65.6 | 73.4 | | 2773 | 2900 | 0.7 | 6.7 | 18.7 | 23.3 | 24.7 | | 10732 | 10897 |
| SnapKV | 49.6 | 66.0 | 74.8 | 80.8 | | 3704 | 3831 | 10.0 | 23.3 | 40.0 | 46.0 | 52.7 | | 13650 | 13815 |
| RKV | 75.0 | 72.2 | 78.4 | 83.6 | | 4229 | 4356 | 14.7 | 32.7 | 43.3 | 55.3 | 61.3 | | 18078 | 18243 |
| FASA | 86.4 | 90.2 | 90.2 | 91.2 | | 2887 | 3014 | 60.7 | 62.0 | 66.3 | 70.0 | 73.2 | | 11735 | 11891 |

Figure 5: Evaluation of FASA on TREC (left) and MATH (right) datasets. The plots show the synergistic effects under varying numbers of selected FCs and different token budgets.

**FASA achieves near-lossless performance under various budgets.** FASA consistently outperforms all baselines across various budgets (Appendix C.1 and 6), preserving contextual integrity even under extreme compression (Table 2). In stark contrast, existing token-eviction methods suffer catastrophic performance degradation; for instance, Quest's accuracy plummets by 13.4% on NarrativeQA, underscoring their inability to retain critical information. Remarkably, under extreme budgets, FASA occasionally surpasses the FKV baseline (e.g., on Mistral-7B). We attribute this phenomenon to the mitigation of attentional distraction from irrelevant tokens. This hypothesis is corroborated by the Oracle baseline, which also outperforms FKV sometimes, thereby validating our frequency-chunk-based framework's efficacy in precisely identifying semantically pivotal regions.

**FASA models complex long-term dependencies.** We simulate a *token-by-token* decoding process wherein the eviction strategy is iteratively applied before token prediction. The fixed-rule approach of

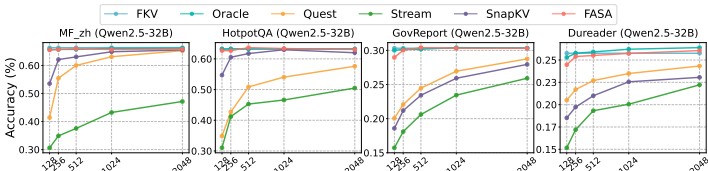

Figure 6: FASA under various token budgets ($N_{tip} = 16$).

Stream (Xiao et al., 2024), which relies on "attention sinks," severely compromises its ability to capture long-range dependencies, leading to a drastic increase in perplexity as shown in Figure 4. Similarly, Quest's coarse, page-level granularity prevents it from adaptively retaining critical, non-contiguous tokens. In contrast, FASA's fine-grained, query-dependent mechanism accurately identifies salient tokens, achieving performance comparable to FKV, even under aggressive compression.

**FASA excels at long-CoT reasoning.** The chain of thought in long-form reasoning is a fragile thread, requiring the preservation of dynamically shifting "thought traces", a thread that prominent baselines

consistently sever. As shown in Table 3, their static compression heuristics, blind to the evolving importance of tokens, lead to a precipitous drop in performance. On R1-Llama, SnapKV's accuracy collapses to 21.6, a stark contrast to the FKV's 72.4, demonstrating a fundamental failure to sustain the very logical dependencies required for reasoning. Conversely, FASA operates with surgical precision. It surpasses not only standard baselines but also R-KV, a highly specialized method for CoT compression. It achieves an impressive 86.4% accuracy on a scant 10% context budget, narrowly trailing the 92.6% FKV upper bound. This feat cements its status as a superior framework, one that can navigate the intricate web of complex reasoning without severing the essential threads of logic.

## 5.3 IN-DEPTH ANALYSIS

Table 4: Compatibility of FASA.

| Budget | 256 | 512 | 1024 | 2048 |
|---|---|---|---|---|
| | | Qasp. | | |
| FASA | 43.7 | 44.0 | 44.7 | 45.7 |
| +PyKV | $44.4_{\uparrow 0.7}$ | $44.5_{\uparrow 0.5}$ | $45.8_{\uparrow 1.1}$ | $45.8_{\uparrow 0.1}$ |
| | | Lcc | | |
| FASA | 61.8 | 63.4 | 64.4 | 64.8 |
| +PyKV | $62.2_{\uparrow 0.4}$ | $63.6_{\uparrow 0.2}$ | $64.7_{\uparrow 0.3}$ | $64.9_{\uparrow 0.1}$ |

Table 5: Ablation on $\mathcal{K}$.

| $\mathcal{K}$ | Token Budget | | | | | AVG. |
|---|---|---|---|---|---|---|
| | 128 | 256 | 512 | 1024 | 2048 | |
| 128 | 42.5 | 43.6 | **44.9** | **45.7** | 45.6 | 44.5 |
| 256 | 42.6 | 43.7 | 44.0 | 44.7 | 45.3 | 44.1 |
| 512 | 41.9 | 43.5 | 43.7 | 44.9 | 45.3 | 43.9 |
| 1024 | 42.2 | 44.2 | 44.3 | 44.7 | 45.0 | 44.1 |

Table 6: Ablation of offline calibration.

| Offline | S-Doc QA | | | M-Doc QA | | |
|---|---|---|---|---|---|---|
| | 2Wiki | Musi | Hqa | Qasp. | MF_en | Nqa |
| Base | 43.7 | 30.2 | 55.8 | 45.3 | 55.6 | 29.9 |
| Nqa | 44.5 | 31.6 | 55.0 | 44.2 | 55.8 | 29.2 |
| Qasp. | 43.0 | 31.0 | 54.1 | 44.0 | 54.6 | 29.1 |
| Musi | 43.8 | 30.8 | 55.1 | 44.8 | 54.6 | 29.6 |
| Self | 43.5 | 30.8 | 55.3 | 43.9 | 54.4 | 29.2 |
| CV | .014 | .012 | .010 | .009 | .011 | .007 |

**Effect on Generation Length.** A neglected aspect of compression methods is the impact on output length. Some compression methods, like H2O, induce generative verbosity, imposing an overlooked computational burden (Table 3). Conversely, others, such as Stream, prematurely terminate generation, which truncates valid reasoning and degrade performance. In contrast, FASA maintains output lengths nearly identical to the FKV while preserving high performance, demonstrating a superior balance.

**Compatiblility of FASA.** By design, FASA is orthogonal to and synergistic with other KV cache optimization paradigms. We demonstrate this by integrating it with PyramidKV (Cai et al., 2025b), which allocates varied budgets across layers. While PyramidKV determines how many tokens to keep per layer, FASA decides which tokens are most critical. As shown in Table 4, this complementary pairing yields consistent performance gains, confirming FASA's high compatibility and modularity.

**Efficiency Analysis.** We assess the efficiency of our two FASA variants. FASA-M's memory savings are particularly pronounced in long sequences, as the KV cache's footprint grows to dominate and dwarf the static memory costs of model parameters and activations. While its CPU-GPU data transfer introduces a slight latency overhead, this can be effectively mitigated by prefetching techniques that asynchronously load the required KV pairs in advance. FASA-C, implemented with Triton (based on Ribar et al. (2024)), delivers substantial inference acceleration. The speedup effect intensifies with longer sequences, achieving up to a 2.56× with $N_{tip} = 16$ under 64K.

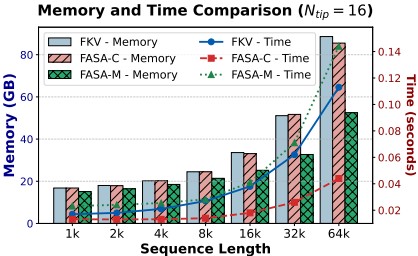

Figure 7: Memory vs. latency ($N_{tip} = 16$).

## 5.4 ABLATION STUDIES

**Robustness to Calibration Window $\mathcal{K}$.** Our method exhibits remarkable robustness to the calibration window size, $\mathcal{K}$. Performance is largely insensitive to $\mathcal{K}$, with smaller $\mathcal{K}$ values often yielding slightly superior results (Table 5). This suggests that due to the inherent sparsity of attention, even a small calibration window provides a sufficiently robust signal to identify the dominant FCs.

**Trade-off between $N_{tip}$ and $N_{fac}$.** The hyperparameters $N_{tip}$ (token selection precision) and $N_{fac}$ (retention budget) govern a trade-off between the fidelity of token identification and the volume of retained context. As depicted in Figure 5, optimal performance can be achieved either with high-precision selection (large $N_{tip}$) and a small budget, or a more lenient selection (small $N_{tip}$) compensated by a larger one. Empirically, on the TREC dataset, we found that using just 10 dominant FCs (15.6% of dimensions) with $N_{fac} = 500$ is sufficient to match the FKV's performance.

**Impact of Offline Calibrated Data.** As shown in Table 6, our method exhibits remarkable robustness to the choice of calibration data. The minimal performance variation across different calibration datasets, as quantified by a low Coefficient of Variation (CV), confirms that our FC detection mechanism is stable and not reliant on a specific calibration source.

## 6    EXTENDING FASA TO NON-RoPE MODELS

In this section, we investigate the generalizability of FASA to non-RoPE architectures. The feasibility of this extension hinges on whether functional sparsity is a universal emergent property induced by alternative PE schemes. We first ascertain the presence of functional sparsity in ALiBi and Partial-RoPE, followed by an empirical evaluation of FASA 's performance within these frameworks.

**Functional Sparsity in ALiBi and Partial-RoPE (MLA)**    We extend our analysis to two widely used PE variants: Attention with Linear Biases (ALiBi) and Partial-RoPE (in Multi-head Latent Attention (MLA)). While ALiBi encodes relative distances using head-specific linear biases added to the attention logits, MLA adopts a decoupled approach where RoPE is applied only to a specific partition of the head dimensions. This strategy allows us to explore sparsity behaviors that go beyond the standard full-RoPE configurations. This strategy enables us to examine sparsity behaviors beyond the typical full-RoPE configurations. As illustrated in Figures 8 and 9, both ALiBi and Partial-RoPE induce functional sparsity at the head dimension level. For ALiBi, heads indexed from 19 to 31 exhibit a regular pattern of functional sparsity, while the remaining dimensions in other heads show high CA scores (approximately 0.7). Therefore, FASA is compatible with ALiBi.

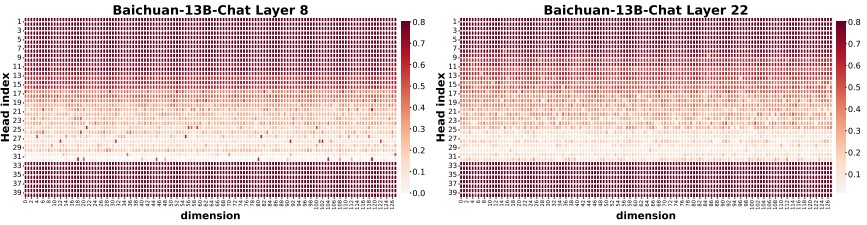

Figure 8: CA Scores Heatmaps of Baichuan-13B-Chat (ALiBi models).

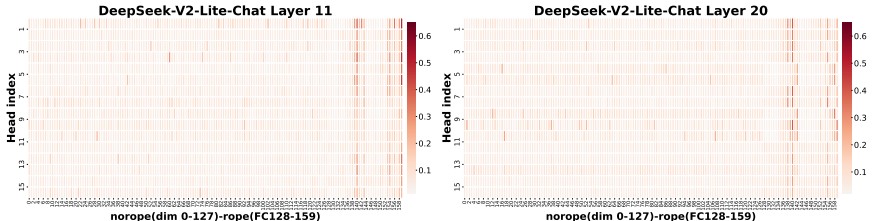

Figure 9: CA Scores Heatmaps of DeepSeek-V2-Lite-Chat (Partial-RoPE models).

**FASA Evaluation on Other PEs**    As demonstrated in Tables 7 and 8, FASA exhibits exceptional generalizability across diverse position encoding architectures, achieving this without incurring any significant performance trade-offs. The results consistently match or surpass those of FKV, *establishing FASA as a robust, high-performance method with broad applicability beyond RoPE.*

Table 7: Performance on Partial-RoPE Models.      Table 8: Performance on ALiBi Models.

| | Qasper | 2Wiki | Multi | Passage_Re | Lcc | Samsum | | Qasper | Lsht | Dureader | Trec | Repobench |
|---|---|---|---|---|---|---|---|---|---|---|---|---|
| FKV | 33.18 | 19.83 | 47.27 | 49.00 | 63.40 | 34.04 | FKV | 9.11 | 24.25 | 23.18 | 23.00 | 17.30 |
| FASA | 33.46 | 20.25 | 46.50 | 48.50 | 62.49 | 32.53 | FASA | 7.80 | 21.25 | 21.70 | 21.50 | 16.46 |

## 7    CONCLUSION

In this work, we addressed the memory footprint and bandwidth introduced by the KV cache in LLMs. Firstly, we cover an intriguing phenomenon: the functional sparsity of FCs. A subset of dominant FCs could show high contextual awareness. Based on this discovery, we introduce FASA, a coarse-to-fine two-stage freamwork. The first stage utilizes the dominant FCs to perform dynamic, query-aware token selection without costly training. Then, the second stage perform focused and precise attention computation on this reduced subset. Our experiments indicate that FASA attains performance nearly on par with full KV even under constrained budgets. The memory- and speed-optimized variants of FASA offers a practical and effective solution for efficient long-context inference.

## ACKNOWLEDGEMENTS

We thank the anonymous reviewers for their insightful comments and suggestions. We also thank the members of the Machine Learning Group at AMAP for their valuable feedback and support throughout the development of this work.

## ETHICS STATEMENT

Our research is focused on enhancing the computational efficiency of Large Language Model (LLM) inference by optimizing KV cache management. The primary positive impact of our work, FASA, is to make large-scale models more accessible, affordable, and environmentally sustainable. By significantly reducing memory and computational overhead, our method can enable researchers and institutions with limited resources to develop and deploy powerful long-context models, thereby fostering broader innovation and democratization in the field of AI.

We acknowledge the dual-use nature of efficiency-enhancing technologies. While our goal is positive, lowering the barrier to running large models could inadvertently make it easier for malicious actors to deploy them for harmful purposes, such as generating misinformation or spam at scale. It is important to note, however, that our work is foundational and does not create new capabilities for generating harmful content; it merely optimizes the performance of existing models.

All experiments were conducted on publicly available benchmarks (LongBench, MATH, AIME) and open-source pre-trained models. We did not use any private, sensitive, or user-generated data. We recognize that the foundation models used in our evaluation may reflect and perpetuate societal biases present in their vast training corpora. Our method operates orthogonally to the challenge of model-level bias and does not address it directly, but we encourage users to be mindful of the inherent limitations of the models they deploy with our technique.

## REPRODUCIBILITY STATEMENT

To ensure the reproducibility of our work, we provide a detailed account of all models, datasets, experimental setups, and evaluation protocols, all of which are publicly available. An overview of the experiments is provided in **Section 5.1**, with more comprehensive details described across several appendices. Specifically, the configurations for all baselines and the detailed hyperparameters for FASA are presented in **Appendix B.1**. The descriptions of all benchmarks and their corresponding evaluation protocols are detailed in **Appendix B.2** and **Appendix B.3**, respectively. Furthermore, the implementation and design choices for FASA are explained in **Appendix B.4**. Finally, the specific algorithms for FASA-M and other core functions are provided in **Appendix D.1** and **Appendix D.3**.

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

# A  INVESTIGATION RESULTS OF DOMINANT FREQUENCY CHUNKS

## A.1  FURTHER GENERALIZATION ON MODEL SCALES AND ARCHITECHTURES

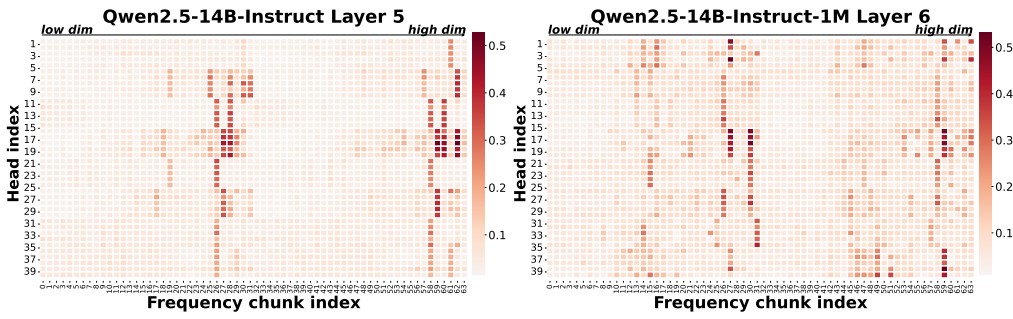

Figure 10: Functional sparsity is maintained on Qwen2.5 series models (Yang et al., 2024). Heatmaps visualize the Mean Contextual Agreement ($\overline{\text{CA}}_{K=256}$) for each Frequency Chunk (FC, x-axis) across all attention heads (y-axis) in a representative layer. We compare the standard **Qwen2.5-14B-Instruct** model (left) with its long-context variant, **Qwen2.5-14B-Instruct-1M** (right), both calibrated on the Qasper dataset. The remarkable similarity between the two heatmaps demonstrates that the functional sparsity of FCs is a robust property, consistently maintained even after long-context fine-tuning.

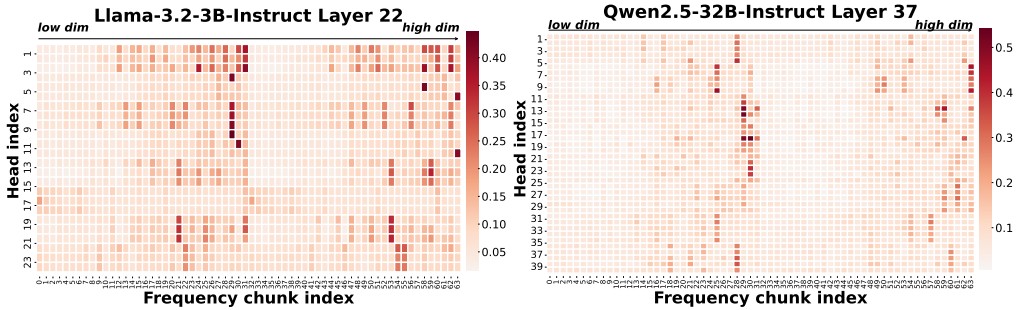

Figure 11: Functional sparsity persists across model scales. Heatmaps show the Mean Contextual Agreement ($\overline{\text{CA}}_{K=256}$) for increasing scale (3B and 32B). The remarkable stability of the dominant FC patterns (bright vertical columns) across these scales demonstrates that functional sparsity is a fundamental and scalable characteristic of RoPE.

**Conclusions:** Our cross-architectural (Figure 10) and cross-scale (Figure 11) analysis reveals a striking finding: the functional sparsity of FCs is a universal and stable property. This powerful evidence suggests that the observed functional hierarchy is not an emergent artifact of a specific model's training dynamics or size, but rather an intrinsic characteristic deeply embedded within the RoPE mechanism itself. The roles of different frequencies appear to be fundamental and pre-determined, providing a robust and predictable foundation for developing model-agnostic efficiency optimizations.

## A.2  TASK-INVARIANCE PROPERTY OF FUNCTIONAL SPARSITY

We find that the saliency of dominant FCs is largely task-agnostic. This property is evidenced by the strong alignment between saliency maps generated for distinct downstream tasks, as shown in Figure 12. Despite the functional differences between question answering (left) and summarization (right), the resulting importance rankings are highly consistent. This indicates that these FCs perform a fundamental role inherent to the model's architecture, rather than one adapted for a specific task.

## A.3  MORE ANALYSIS RESULTS

**Functional Sparsity across Layers.**  While the principle of functional sparsity is universal, the specific set of dominant FCs is far from static in Figure 13; instead, it exhibits a high degree of

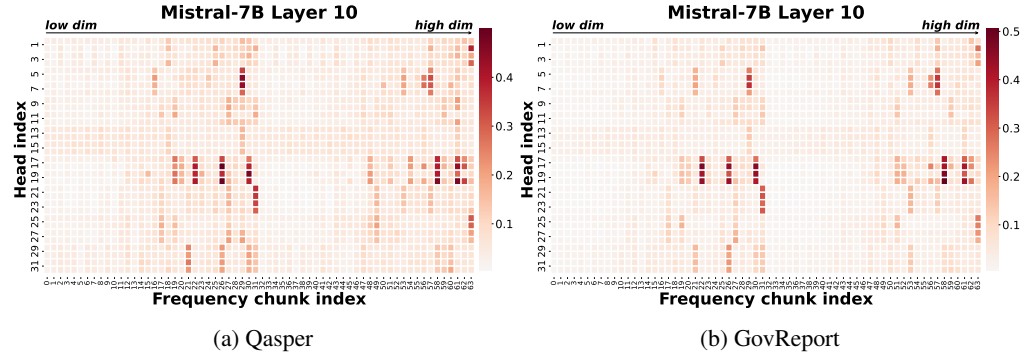

(a) Qasper       (b) GovReport

Figure 12: Heatmaps of agreement score ($\overline{\text{CA}}$, $K = 256$) across attention heads for the Qasper (Left) and GovReport (Right) from LongBench-V1 (Bai et al. (2024)) on Mistral-7B-Instruct-v0.3.

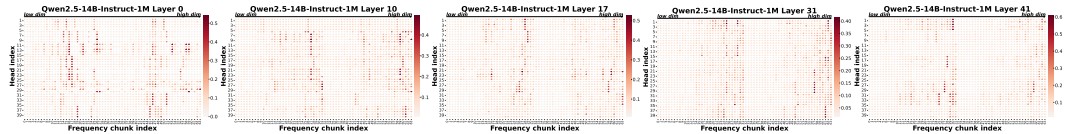

Figure 13: Heatmaps of agreement score ($\overline{\text{CA}}$, $K = 256$) across different layers.

specialization across both model depth and individual attention heads. This dynamic behavior reveals a sophisticated division of labor within the transformer architecture.

## A.4 QUANTITATIVE EVIDENCE ON SPARSITY & UNIVERSALITY & TASK- INVARIANCE

Table 9: The ratio of dominant FCs and non-dominant FCs.

| Type of FC | Dominant FCs (%) | Non-Dom FCs (%) |
|---|---|---|
| **Model** | **CA scores $> 0.4$** | **CA score $< 0.15$** |
| Llama-3.2-3B | 0.54 | 89.6 |
| Meta-Llama-3.1-8B | 0.68 | 89.6 |
| Mistral-7B-v0.3 | 0.68 | 92.7 |
| Qwen2.5-7B | 0.17 | 95.5 |
| Qwen2.5-14B | 0.27 | 94.7 |
| Qwen2.5-14B-1M | 0.65 | 90.5 |
| Qwen2.5-32B-Instruct | 0.52 | 91.2 |
| R1-Distill-Llama-8B | 0.79 | 89.5 |
| R1-Distill-Qwen-14B | 0.76 | 90.2 |
| R1-Distill-Qwen-32B | 0.67 | 90.9 |

Table 10: Cross-task overlap matrix of dominant FCs (%). Each sub-table shows the percentage of intersection between dominant FCs identified on a "row" dataset and a "column" dataset.

| Model | Overlap of Dom-FCs | Qasper | Gov_Report | Musique | Narrativeqa | 2Wikimqa | Avg. |
|---|---|---|---|---|---|---|---|
| | Qasper | 100.00 | 75.90 | 82.30 | 70.50 | 83.20 | 82.38 |
| | Gov_Report | 75.90 | 100.00 | 82.10 | 70.80 | 81.90 | 82.14 |
| Llama-3.2-3B | Musique | 82.30 | 82.10 | 100.00 | 73.60 | 96.50 | 86.90 |
| | Narrativeqa | 70.50 | 70.80 | 73.60 | 100.00 | 73.10 | 77.60 |
| | 2Wikimqa | 83.20 | 81.90 | 96.50 | 73.10 | 100.00 | 86.94 |
| | Qasper | 100.00 | 71.10 | 77.10 | 67.30 | 77.00 | 78.50 |
| | Gov_report | 71.10 | 100.00 | 79.40 | 65.50 | 78.90 | 78.98 |
| Mistral-7B | Musique | 77.10 | 79.40 | 100.00 | 67.80 | 97.90 | 84.44 |
| | Narrativeqa | 67.30 | 65.50 | 67.80 | 100.00 | 67.30 | 73.58 |
| | 2Wikimqa | 77.00 | 78.90 | 97.90 | 67.30 | 100.00 | 84.22 |
| | Qasper | 100.00 | 70.60 | 80.90 | 68.70 | 81.30 | 80.30 |
| | Gov_Report | 70.60 | 100.00 | 79.40 | 68.20 | 78.70 | 79.38 |
| Qwen2.5-7B | Musique | 80.90 | 79.40 | 100.00 | 71.70 | 96.60 | 85.72 |
| | Narrativeqa | 68.70 | 68.20 | 71.70 | 100.00 | 71.10 | 75.94 |
| | 2Wikimqa | 81.30 | 78.70 | 96.60 | 71.10 | 100.00 | 85.54 |
| | Qasper | 100.00 | 69.20 | 84.30 | 71.80 | 84.50 | 81.96 |
| | Gov_Report | 69.20 | 100.00 | 75.00 | 67.60 | 74.80 | 77.32 |
| Qwen2.5-14B | Musique | 84.30 | 75.00 | 100.00 | 74.30 | 98.40 | 86.40 |
| | Narrativeqa | 71.80 | 67.60 | 74.30 | 100.00 | 73.90 | 77.52 |
| | 2Wikimqa | 84.50 | 74.80 | 98.40 | 73.90 | 100.00 | 86.32 |

Table 11: Predictive distribution of dominant FCs across different attention score ranges.

| | | Prediction accuracy across varying attention scale ranges | | | | |
|---|---|---|---|---|---|---|
| Model | Type of FCs | Top 20% | Top 20-40% | Top 40-60% | Top 60-80% | Top 80-100% |
| Llama-3.2-3B-Instruct | Dom | 82.4* | 79.1 | 72.1 | 59.2 | 44.9 |
| | Non-dom | 4.6 | 5.3 | 5.3 | 5.4 | 5.4 |
| Mistral-7B-Instruct-v0.3 | Dom | 81.1 | 80.7 | 78.7 | 72.5 | 56.4 |
| | Non-dom | 3.6 | 4.2 | 4.9 | 4.4 | 4.5 |
| Qwen2.5-7B-Instruct | Dom | 81.9 | 82.4 | 76.9 | 63.7 | 49.3 |
| | Non-dom | 6.1 | 5.7 | 5.4 | 5.6 | 5.5 |
| Qwen2.5-14B-Instruct | Dom | 74.3 | 66.4 | 56.6 | 44.9 | 34.7 |
| | Non-dom | 4.1 | 4.6 | 4.5 | 4.9 | 4.9 |

# B EXPERIMENTS DETAILS

## B.1 EXPERIMENT CONFIGURATIONS.

**Baseline Configurations.** As FASA is designed to optimize the decode phase, we forgo any KV cache optimizations during prefilling for all methods under evaluation. This experimental design isolates the performance impact of decode-stage acceleration, ensuring that our comparisons are direct and fair. For all baselines, we adopted configurations that are either standard in their original papers or represent a fair and strong setup for comparison.

- **Oracle**: serves as an oracle baseline to demonstrate the upper-bound performance of Top-k sparse attention. This method operates under the ideal assumption that the k most important KV tokens for each query can be identified perfectly and at no computational cost. Consequently, a given token budget directly corresponds to this optimal Top-k set.
- **Stream** (Xiao et al., 2024): This method is based on the "attention sink" phenomenon, preserving a fixed number of initial tokens and a sliding window of recent tokens. Following its standard setup, we set the initial "start_size" to 8 and the "recent_size" to "budget - 8".
- **SnapKV** (Li et al., 2024): SnapKV estimates token importance based on accumulated attention scores within a observation window during prefilling. We adopted its "maxpool" strategy with a window size of 32 and a kernel size of 7. As its original design performs a one-time filtering, it is

not directly suited for long-generation tasks. We therefore adapted it, following the methodology in (Cai et al., 2025a), by re-applying the filtering mechanism every $n$ generated tokens.
- **Quest** (Tang et al., 2024): Quest organizes the KV cache into pages and retrieves them based on a coarse-grained query-page similarity. We set the page size to 16, a value reported as near-optimal, to balance the trade-off between retrieval granularity and overhead.
- **RKV** (Cai et al., 2025a): RKV is a state-of-the-art method for reasoning tasks that also employs a retrieval mechanism. We set its core hyperparameter $\lambda$, which balances between recent and important tokens, to 0.1 as recommended for optimal performance.

**FASA Configurations.** Our configuration for FASA is designed for both effectiveness and practical efficiency. Unless otherwise specified, the following setup was used across all experiments.

- **Dominant FC Identification:** A core principle of FASA is that the set of dominant FCs is a universal, task-agnostic property of the model architecture itself. Consequently, these indices ($\mathcal{I}_{dom}$) can be determined via a highly efficient, one-time offline calibration. For our **LongBench** experiments, this calibration was performed on just a single data sample from the Qasper dataset. We found this minimal setup to be remarkably robust, as the generated response provides sufficient signal to identify the dominant FCs. The universality of these calibrated indices is empirically validated by FASA's strong performance across diverse tasks, from summarization to code completion. For **Long-CoT reasoning**, a similar single-instance calibration was performed on a question from the MATH500 dataset.
- **Hyperparameter Settings:** For architectural simplicity and to maximize computational parallelism, we employ a uniform configuration across all heads and layers. The number of dominant FCs to retain, denoted as $N_{\text{tip}}$, was consistently set to 16. This choice represents a balance between preserving sufficient contextual information and maximizing computational.
- **Task Configurations:** We configured the maximum sequence length to 32k for the AIME24 benchmark, reflecting its higher reasoning complexity, and to 16k for MATH500. For the LongBench benchmark, we set the maximum prompt length to 127.5k for Llama3/Qwen2.5 series models and 31.5k for Mistral-7B-Instruct-v0.2.

## B.2 BENCHMARK DETAILS

**LongBench (Bai et al., 2024)** is a comprehensive, multi-task benchmark designed to evaluate the long-context understanding capabilities of Large Language (Wang et al., 2024; 2025b). It comprises a diverse set of tasks, including single-document QA, multi-document QA, summarization, few-shot learning, synthetic tasks, and code completion. In our experiments, we report the average performance across all relevant tasks to provide a holistic measure of a model's ability to process and reason over extended contexts, with sequence lengths ranging from 4K to over 100K tokens.

**MATH500  (Hendrycks et al., 2021)** is a challenging benchmark for evaluating mathematical reasoning. It consists of 12,500 problems sourced from high school math competitions, spanning subjects like Algebra, Geometry, Number Theory, and Precalculus. Each problem is accompanied by a step-by-step solution, making it highly suitable for assessing CoT reasoning capabilities. We utilize the MATH500 subset for our long-CoT generation experiments, where models must produce detailed reasoning chains to arrive at the final answer.

**AIME (MAA, 2024)** represents a significant step-up in reasoning complexity compared to the MATH dataset. It consists of problems from the AIME competition, which are known for their non-routine, multi-step solutions requiring deep mathematical insight and creativity (Li et al., 2025; Dai et al., 2026). These problems serve as a stress test for a model's most advanced reasoning and long-chain generation abilities (Xiong et al., 2025). Following standard practice, we evaluate performance using the pass@k metric, specifically reporting pass@1 based on 16 generated responses per question.

**C4** (Raffel et al., 2019) is a massive, general-domain English text dataset derived from the Common Crawl web scrape. The "clean" version is created by applying a series of heuristics to filter out boilerplate content, code, and offensive language, resulting in a high-quality, natural language corpus.

**PG19** (Rae et al., 2019) is a long-form text dataset derived from books in the Project Gutenberg library. It is specifically curated for evaluating long-range sequence modeling. Each example in the

dataset is a full book text, making it an ideal benchmark for assessing a model's ability to handle and maintain coherence over very long dependencies, often exceeding the context windows of LLMs.

**WikiText**(Merity et al., 2017) is a large-scale language modeling corpus sourced from high-quality "Good" and "Featured" articles on Wikipedia. Unlike raw web text, WikiText is well-formatted, grammatically correct, and retains its original punctuation and case. It is split into training, validation, and test sets at the article level.

### B.3 EVALUATION PROTOCOLS

To provide a comprehensive and rigorous assessment of model performance, we employ a set of standard metrics tailored to each evaluation paradigm.

**Long-Context Understanding (LongBench).** For the diverse tasks within the LongBench (Bai et al., 2024), we follow its official evaluation protocol. Specifically, we use:

- **f1 score** for question-answering tasks.
- **rouge_score** for summarization tasks.
- **code_sim_score** for code completion tasks.

The final reported score for LongBench is the average performance across all constituent tasks.

**Long-Sequence Modeling.** To evaluate a model's ability to maintain generative fidelity over long dependencies, we use perplexity (PPL). Perplexity measures how well a probability model predicts a sample. For a sequence of tokens $W = (w_1, w_2, \ldots, w_N)$, PPL is defined as the exponential of the average negative log-likelihood in Equation 9. A lower PPL indicates a better model, as it signifies higher confidence and accuracy in predicting the next token.

$$\text{PPL}(W) = \exp\left(-\frac{1}{N}\sum_{i=1}^{N}\log P(w_i|w_{<i})\right) \tag{9}$$

**Long CoT Reasoning.** For complex mathematical reasoning tasks such as MATH500 and AIME2024, we evaluate the model's performance in a long-generation setting. This paradigm is distinct from conventional long-context understanding tasks. Instead of processing a long static input, the model must maintain logical coherence and track thought traces across an extended, auto-regressive generation process to produce the correct final answer. Performance is reported as pass@1 (Dai et al., 2026).

- For MATH500, we report pass@1, where a single generation is sampled for each problem.
- For AIME2024, which features more challenging problems, we also report pass@1, but the result is determined by checking if at least one correct answer exists within $k = 16$ independent generations for each question. This sampling strategy is standard for estimating performance on complex reasoning benchmarks.

### B.4 IMPLEMENT DETAILS

**Implementation Details** Our implementation of FASA is built upon the HuggingFace Transformers library (Wolf et al., 2020). We employ a non-invasive monkey patching approach to integrate our logic. Specifically, we intercept the forward pass of the FlashAttention2 class within the model's modeling.py file. The core of our method resides in two components. First, leveraging the universal nature of dominant FCs, their pre-computed indices are stored in a globally accessible dictionary, shared across all layers and heads. Second, the Token Importance Prediction (TIP) logic, which performs the critical token selection, is encapsulated within our core_module_with_padding function. A key advantage of our design is its simplicity and minimal intrusion. The integration requires inserting just a single line of code, the token selection logic, into the original attention function, making FASA easy to deploy and adapt. This minimal intrusion makes FASA highly portable and easy to adapt. The corresponding pseudocode is provided in Figure 14.

```
bsz, q_len, _ = hidden_states.size()
cos, sin = position_embeddings
query_states, key_states = apply_rotary_pos_emb(query_states, key_states, cos, sin)
####################################################################
#token selection in TIP
if query_states.shape[2] == 1: # for deocoding stage
    key_states,value_states = core_module_with_padding(query_states,\
        key_states,value_states,self.layer_idx,budget,records)
####################################################################
query_states = query_states.transpose(1, 2)
key_states = key_states.transpose(1, 2)
value_states = value_states.transpose(1, 2)
attn_output = _flash_attention_forward(
query_states,
key_states,
value_states,
attention_mask,
q_len,
dropout=dropout_rate,
sliding_window=getattr(self, "sliding_window", None),
use_top_left_mask=self._flash_attn_uses_top_left_mask,
is_causal=self.is_causal,
)
attn_output = attn_output.reshape(bsz, q_len, -1).contiguous()
attn_output = self.o_proj(attn_output)
return attn_output, attn_weights, past_key_value
```

Figure 14: The FASA Pipeline: An Efficient, FlashAttention-Compatible Approach. The algorithm details our two-stage process. A key design feature is that the FAC stage seamlessly integrates with the standard FlashAttention API, leveraging its performance while enabling sparse computation.

## C  ADDITIONAL EXPERIMENTAL RESULTS

### C.1  PERFORMANCE ANALYSIS ON DIFFERENT BUDGETS

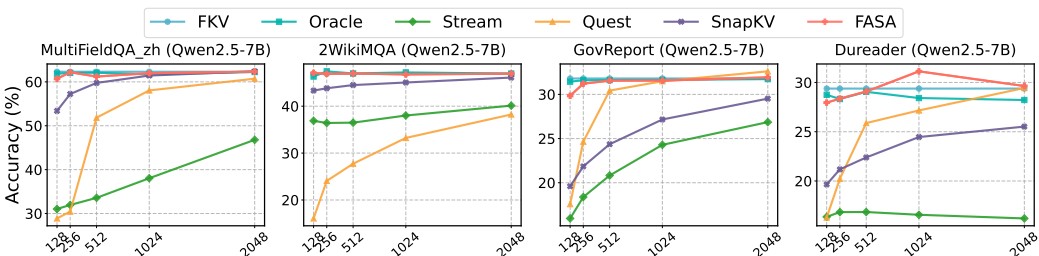

Figure 15: FASA on Qwen2.5-7B-Instruct under various token budgets ($N_{\text{tip}} = 16$).

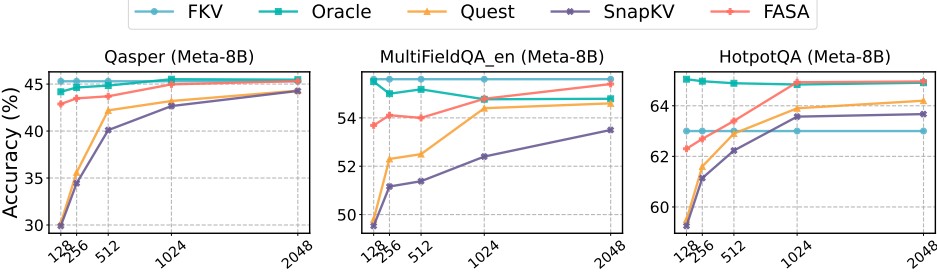

Figure 16: FASA on Meta-3.1-Llama-8B-Instruct under various token budgets ($N_{\text{tip}} = 16$).

**Comparison with Low-Rank Methods**  A closely related work to FASA is SparQ (Ribar et al., 2024), which also performs a form of dimension selection. SparQ operates on the heuristic that high-magnitude dimensions in a query vector are the most indicative of importance, and thus selects corresponding key dimensions as a proxy for token prediction. However, as our experiments in Figure 17 demonstrate, this heuristic proves to be a poor substitute for true contextual awareness.

Under a constrained budget of 256 tokens, SparQ's performance collapses, indicating its inability to reliably identify critical tokens based solely on query magnitudes. Furthermore, from an efficiency standpoint, SparQ incurs significant overhead as it must re-evaluate high-magnitude dimensions for every new query. In stark contrast, FASA leverages a one-time, offline calibration, making its per-token inference cost substantially lower.

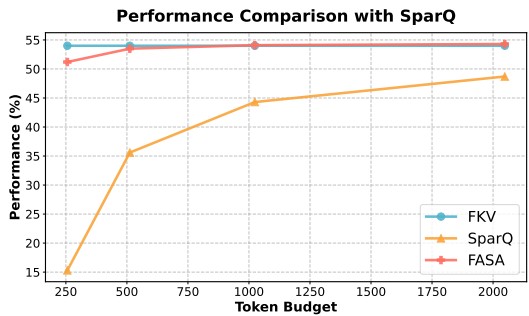

Figure 17: Comparision with SparQ on LongBench.

## D    DISCUSSION ON FASA

### D.1    VARIANTS OF FASA

**FASA-M (Memory-Optimized)**    The memory-optimized variant, FASA-M, is specifically engineered for scenarios with constrained GPU memory, such as consumer-grade hardware. As detailed in Algorithm 2, its core strategy is to minimize the on-GPU memory footprint by strategically keeping only the most essential data on the GPU.

Specifically, only the dominant parts of the Key cache ($C_{key}^{dom}$), which are required for the initial token importance prediction, are retained in GPU memory. The non-dominant parts of the Key cache ($C_{key}^{nondom}$) and the entire Value cache ($C_{val}$) are offloaded to and managed in the much larger CPU memory. During the Focused Attention Computation (FAC) stage, once the critical token indices ($\mathcal{T}_t$) are identified, only the small, required subsets of the non-dominant key and value caches are transferred from the CPU to the GPU for the final attention calculation. This "just-in-time" data transfer ensures that the GPU memory is primarily occupied by the most critical components, leading to substantial memory savings.

**Memory Footprint Analysis**    The GPU memory footprint of the KV cache in FASA-M can be formulated as follows. Let $L$ be the total sequence length, $b$ the token budget, $d$ the model's hidden dimension, and $N_{layers}$ the number of layers. Let $d_{dom}$ be the dimension of the dominant FCs and $d_{nondom}$ be the dimension of the non-dominant FCs ($d = d_{dom} + d_{nondom}$). The memory occupied by the KV cache on the GPU is:

$$\text{Mem}_{\text{GPU}} \approx N_{layers} \times \left( \underbrace{L \times d_{dom}}_{\text{Dominant Keys}} + \underbrace{b \times d_{nondom}}_{\text{Non-dominant Keys}} + \underbrace{b \times d}_{\text{Values}} \right) \times \text{bytes\_per\_param} \qquad (10)$$

Compared to a full KV cache, which occupies $N_{layers} \times L \times 2d \times \text{bytes\_per\_param}$, FASA-M significantly reduces the memory burden, especially when the non-dominant and value components constitute a large portion of the cache. For instance, if $d_{dom}$ is 25% of $d$ and the budget $b$ is 10% of $L$, the memory savings can be substantial, approaching an $8\times$ reduction in typical configurations.

### D.2    DESIGN CHOICES

- **On the Role of FC-Scores: A Proxy for Ranking, Not a Substitute for Attention.** A crucial design principle we validated is that our FC-based scores ($\mathbf{S}_t^{l,h}$) are not calibrated to function as direct attention weights. Although they provide a remarkably accurate relative ranking of token

importance, their direct substitution for attention probabilities leads to a catastrophic performance degradation. This reveals their fundamental role as a selector—a mechanism to identify salient tokens rather than an approximator of the final attention distribution.

- **On the Indivisibility of Frequency Chunks.** We investigated whether individual dimensions could serve as selection units, and the answer is a definitive no. A pipeline based on selecting "dominant dimensions" suffers a catastrophic performance degradation. This empirically validates that the Frequency Chunk (FC) is an indivisible functional unit for this process. This principle is not coincidental but is a direct corollary of RoPE's core mechanism, which encodes position by applying rotations to coupled pairs of dimensions. Disrupting these pairs severs the positional encoding, leading to model failure.

In summary, these two findings underscore two core design principles of FASA. First, an efficient proxy for token importance does not necessarily serve as a valid substitute for attention weights. Second, any optimization for RoPE-based models must respect the inherent coupling of dimension pairs, treating the Frequency Chunk as an indivisible functional unit.

## D.3 ALGORITHM ON FASA

See the algorithm of offline calibration in Algorithm 1; see the algorithm of FASA-M in Algorithm 2.

---

**Algorithm 1:** Offline Calibration for Dominant FCs

**Input:** A calibration dataset $\Omega$; number of dominant FCs to select $k$.
**Output:** The set of dominant FC indices, $\mathcal{I}_{dom}$.

```
// Stage 1:  Collect Contextual Agreement (CA) scores
```
Initialize an empty map $M$ to store CA scores for each $(l, h, i)$ triplet
**foreach** *example in $\Omega$* **do**
    **foreach** *token generation step $t$* **do**
        **foreach** *layer $l$* **do**
            **foreach** *head $h$* **do**
                Compute full attention scores $\boldsymbol{\alpha}_{l,h}(\mathbf{q}_t, \mathbf{K}_{1:t})$
                **foreach** *FC index $i$* **do**
                    Compute single-FC scores $\boldsymbol{\alpha}_{l,h}^{(i)}(\mathbf{q}_t, \mathbf{K}_{1:t})$
                    Calculate the CA score $\mathrm{CA}_{\mathcal{K}}^{l,h,i}$ using Eq. 4
                    Store $\mathrm{CA}_{\mathcal{K}}^{l,h,i}$ in $M[l][h][i]$
                **end**
            **end**
        **end**
    **end**
**end**
```
// Stage 2:  Select Dominant FCs
```
Initialize an empty map $\overline{M}$ for mean CA scores
**foreach** $(l, h, i)$ *in $M$* **do**
    $\overline{M}[l][h][i] \leftarrow \mathrm{Mean}(M[l][h][i])$
**end**
$\mathcal{I}_{dom} \leftarrow \mathrm{TopK\text{-}Indices}(\overline{M}, k)$ `// Select top-k indices based on` $\overline{\mathrm{CA}}$
**return** $\mathcal{I}_{dom}$

---

## E LLM USAGE

During the preparation of this manuscript, we utilized the AI-based language model ChatGPT, developed by OpenAI. Its use was strictly limited to language refinement, including grammar correction, stylistic enhancement, and rephrasing for clarity. All scientific concepts, experimental

---

**Algorithm 2:** Inference with FASA-M (Memory-Optimized Variant)

---

**Input:** Current query $\mathbf{q}_t$; Current key $\mathbf{k}_t$; Current value $\mathbf{v}_t$
Dominant FC indices $\mathcal{I}_{dom}$
Token budget $b$
Past KV cache: $C_{key}^{dom}$ (GPU), $C_{key}^{nondom}$ (CPU), $C_{val}$ (CPU)
**Output:** Next hidden state $\mathbf{h}_{t+1}$
Updated KV cache: $C_{key}^{dom}, C_{key}^{nondom}, C_{val}$

```
// Stage 1:  Token Importance Prediction (TIP)
// Split key by dominant FCs
```
$\mathbf{k}_t^{dom}, \mathbf{k}_t^{nondom} \leftarrow \text{Split}(\mathbf{k}_t, \mathcal{I}_{dom})$
```
// Select corresponding query dimensions
```
$\mathbf{q}_t^{dom} \leftarrow \text{Select}(\mathbf{q}_t, \mathcal{I}_{dom})$
$K_{1:t}^{dom} \leftarrow \text{UpdateCache}(C_{key}^{dom}, \mathbf{k}_t^{dom})$
```
// Approximate scores using dominant parts
```
$\hat{\mathbf{S}}_t \leftarrow \mathbf{q}_t^{dom}(K_{1:t}^{dom})^\top$
```
// Identify indices of b most salient tokens
```
$\mathcal{T}_t \leftarrow \text{TopK-Indices}(\hat{\mathbf{S}}_t, b)$

```
// Stage 2:  Focused Attention Computation (FAC)
// Select dominant key parts on GPU
```
$K_{\mathcal{T}_t}^{dom} \leftarrow \text{SelectTokens}(K_{1:t}^{dom}, \mathcal{T}_t)$
```
// Update non-dominant cache on CPU
```
$C_{key}^{nondom} \leftarrow \text{UpdateCache}(C_{key}^{nondom}, \mathbf{k}_t^{nondom})$
$K_{1:t}^{nondom} \leftarrow \text{LoadFromCPU}(C_{key}^{nondom})$
```
// Select non-dominant key parts on CPU
```
$K_{\mathcal{T}_t}^{nondom} \leftarrow \text{SelectTokens}(K_{1:t}^{nondom}, \mathcal{T}_t)$
```
// Update value cache on CPU
```
$C_{val} \leftarrow \text{UpdateCache}(C_{val}, \mathbf{v}_t)$
$V_{1:t} \leftarrow \text{LoadFromCPU}(C_{val})$
```
// Select values on CPU
```
$V_{\mathcal{T}_t} \leftarrow \text{SelectTokens}(V_{1:t}, \mathcal{T}_t)$

```
// Offload required non-dominant keys to GPU
```
$K_{\mathcal{T}_t}^{nondom} \leftarrow \text{TransferToGPU}(K_{\mathcal{T}_t}^{nondom})$
```
// Offload required values to GPU
```
$V_{\mathcal{T}_t} \leftarrow \text{TransferToGPU}(V_{\mathcal{T}_t})$
```
// Reconstruct full keys for selected tokens
```
$K_{\mathcal{T}_t} \leftarrow \text{Combine}(K_{\mathcal{T}_t}^{dom}, K_{\mathcal{T}_t}^{nondom}, \mathcal{I}_{dom})$
```
// Compute full attention on the subset
```
$\boldsymbol{\alpha}_{\text{fac}} \leftarrow \text{Softmax}(\mathbf{q}_t K_{\mathcal{T}_t}^\top / \sqrt{d_k})$
$\mathbf{h}_{t+1} \leftarrow W_O(\boldsymbol{\alpha}_{\text{fac}} V_{\mathcal{T}_t})$
**return** $\mathbf{h}_{t+1}$ *and updated caches*

---

designs, data analyses, and conclusions presented herein are the original work of the authors and were conceived and executed without any substantive contribution from the language model.

