# OpenReview forum: "FASA: FREQUENCY-AWARE SPARSE ATTENTION"
_ICLR.cc/2026/Conference — ICLR 2026 Poster_

### Official Review · Reviewer_UWTV · 2025-10-18

**Soundness:** 4
**Presentation:** 4
**Contribution:** 4
**Rating:** 8
**Confidence:** 4

**Summary:**

This paper proposes FASA (Frequency-Aware Sparse Attention), a training-free, query-aware framework for token eviction by dynamically predicting token importance in LLMs. The authors identify functional sparsity in frequency chunks (FCs) within the RoPE positional encoding, observing that only a small subset (“dominant FCs”) contributes significantly to contextual awareness. Leveraging this insight, FASA consists of two stages:
1. Token Importance Prediction (TIP): Uses dominant FCs to predict salient tokens.
2. Focused Attention Computation (FAC): Performs full attention only on these tokens.
Two variants are proposed—FASA-M (memory optimized) and FASA-C (computation optimized)—achieving up to 2.56× speedup while maintaining near full-KV accuracy across LongBench-V1, MATH500, and AIME24.

**Strengths:**

1. The discovery of functional sparsity at the frequency-chunk level in RoPE is both novel and conceptually insightful. The paper is clearly written and well-structured.
2. FASA’s training-free, model-agnostic, and task-independent design enables straightforward integration with existing LLMs, greatly enhancing its practical deployability.
3. The method demonstrates strong performance

**Weaknesses:**

1. Although FASA reduces computation by selective token attention, it still retains the full KV cache, meaning memory consumption remains a potential bottleneck for extremely long contexts.

**Questions:**

1. In Figure 10, it would strengthen the universality claim to evaluate across a broader range of popular models.
2. I would appreciate a more intuitive explanation of dominant frequency chunks (FCs). The Contextual Agreement (CA) score measures alignment in token ranking between the full-head and single-FC attention scores. A high CA score indicates that the FC captures a similar ranking of token importance. However, since CA reflects order rather than magnitude, it is possible that some non-dominant FCs might exhibit smaller agreement but larger attention scale. Showing the distribution of CA scores for selected dominant FCs could help demonstrate that they accurately capture both ranking and influence on token importance.

---

> ### Author Response · Authors · 2025-11-21
> **Response to Reviewer UWTV 1/2**
>
> ## **Response to W1**
>
> Thank you for raising this excellent point regarding memory efficiency, which is a crucial aspect of our work. This is precisely the challenge that motivated us to design FASA-M, a memory-optimized variant detailed in Section 4.3 of our paper. It is built upon a hybrid memory strategy:
>
> - **Hierarchical Storage:** It stores the vast majority of "non-dominant" dimensions on the CPU, while keeping only the critical "dominant" FCs on the GPU.
> - **On-Demand Loading:** During each token generation step, only the non-dominant dimensions for a small subset of important tokens are loaded from the CPU to the GPU as needed.
>
> This hierarchical storage and on-demand loading mechanism is specifically designed to minimize the peak GPU memory footprint (see Figure 7). We hope this explanation clarifies how FASA-M effectively addresses your concerns.
>
> ## **Response to Q1**
>
> We are grateful for the reviewer's highly constructive feedback on strengthening our universality claim. Your suggestion prompted us to move beyond qualitative demonstrations and substantiate our analysis with rigorous quantitative evidence.
>
> To that end, we have expanded our quantitative analysis to a broader range of popular models to validate the property of functional sparsity from **three key perspectives: Sparsity, Universality, and Task-Invariance.** We believe this comprehensive analysis directly addresses your concern.
>
> ### **Quantitative analysis about the property of Sparsity, Universality, & Task-Invariance**
>
> | Type of FC   | Dominant FCs (%) (CA > 0.4) | Non-Dom FCs (%) (CA < 0.15) |
> |------------------------------|-----------------------------|-----------------------------|
> | **Model**   | **CA scores > 0.4**   | **CA score < 0.15**         |
> | Llama-3.2-3B   | 0.54  | 89.6  |
> | Meta-Llama-3.1-8B    | 0.68  | 89.6  |
> | Mistral-7B-v0.3    | 0.68   | 92.7  |
> | Qwen2.5-7B    | 0.17 | 95.5  |
> | Qwen2.5-14B   | 0.27 | 94.7  |
> | Qwen2.5-14B-1M   | 0.65 | 90.5 |
> | Qwen2.5-32B| 0.52 | 91.2  |
> | R1-Distill-Llama-8B   | 0.79   | 89.5 |
> | R1-Distill-Qwen-14B | 0.76  | 90.2   |
> | R1-Distill-Qwen-32B   | 0.67   | 90.9   |
>
> Table1: The ratio of dominant FCs (CA scores > 0.4) and non-dominant FCs (CA score < 0.15) among different models.
>
> - **Sparsity:**  We quantitatively analyzed the proportion of dominant FCs (defined as CA > 0.4). Our findings reveal a stark imbalance: **dominant FCs consistently account for less than 1% of the total, while non-dominant FCs (CA < 0.15, for example) constitute a vast majority, comprising 90% or more.** This extreme sparsity provides strong empirical validation for our claim of sparsity.
>
> - **Universality:**  This sparsity pattern holds universally. We confirmed its existence across different architectures (Llama, Qwen, Mistral, R1 models) and scales (3B to 32B), which strongly supports our universality claim in Table 1.
>
> | Model  | Overlap of dom-FCs | Qasper | Gov_Report | Musique | Narrativeqa | 2Wikimqa | Avg.   |
> |-------------------|-------------------|--------|------------|---------|-------------|----------|--------|
> | **Llama-3.2-3B** | Qasper| 100.00 | 75.90      | 82.30   | 70.50       | 83.20    | 82.38  |
> |  | Gov_Report  | 75.90  | 100.00 | 82.10   | 70.80       | 81.90    | 82.14  |
> |   | Musique   | 82.30  | 82.10 | 100.00  | 73.60       | 96.50    | 86.90  |
> |    | Narrativeqa  | 70.50  | 70.80 | 73.60   | 100.00      | 73.10    | 77.60  |
> |  | 2Wikimqa | 83.20  | 81.90| 96.50 | 73.10 | 100.00   | 86.94  |
> | **Mistral-7B** | Qasper   | 100.00 | 71.10 | 77.10   | 67.30 | 77.00    | 78.50  |
> |  | Gov_Report | 71.10  | 100.00  | 79.40   | 65.50  | 78.90| 78.98  |
> |  | Musique   | 77.10  | 79.40 | 100.00  | 67.80 | 97.90 | 84.44  |
> |   | Narrativeqa  | 67.30  | 65.50 | 67.80   | 100.00 | 67.30  | 73.58 |
> |  | 2Wikimqa | 77.00  | 78.90  | 97.90   | 67.30 | 100.00   | 84.22  |
> | **Qwen2.5-7B**| Qasper| 100.00 | 70.60| 80.90| 68.70 | 81.30 | 80.30  |
> |  | Gov_Report| 70.60| 100.00 | 79.40| 68.20  | 78.70 | 79.38  |
> |  | Musique   | 80.90  | 79.40 | 100.00  | 71.70  | 96.60 | 85.72  |
> |  | Narrativeqa  | 68.70  | 68.20| 71.70| 100.00 | 71.10| 75.94  |
> | | 2Wikimqa   | 81.30  | 78.70  | 96.60 | 71.10 | 100.00| 85.54  |
>
> Table2: Cross-task overlap matrix of dominant FCs (%).
>
> - **Task-Invariance:** we measured the overlap between dominant FC sets identified using calibration data from diverse downstream tasks. Our analysis reveals **a remarkably high degree of overlap, consistently exceeding 70% across all tested models and task pairings.**
> This high level of consistency provides strong evidence that the set of dominant FCs is **not merely an artifact of the calibration task but is rather an intrinsic property of the model's architecture.** This finding is further corroborated by its generalizability, as the pattern holds true across different task pairings.
>
> **Collectively, these findings demonstrate that our core claim holds true across a diverse range of models.**

---

> ### Author Response · Authors · 2025-11-21
> **Response to Reviewer UWTV 2/2**
>
> ## **Response to Q2**
>
> Thank you for your valuable and insightful feedback. We agree that a deeper analysis of the CA score distribution and a more intuitive explanation of the effect of dominant FCs are crucial for a complete understanding.
>
>  In response, we conducted an additional analysis specifically designed to examine the predicting accuracy of dominant and non-dominant FCs on tokens of varying attention magnitudes (i.e., importance levels).
>
>  **Predictive Accuracy Across Attention Score Ranges:**  We first identified the top 256 tokens with the highest attention scores.  This set of tokens was then stratified into five quintiles based on attention score magnitude: Top 0–20%, Top 20–40%, Top 40–60%, Top 60–80%, and Top 80–100%. For each quintile, we measured the predictive accuracy of two distinct FC groups: Dominant FCs (defined as CA > 0.6) and Non-dominant FCs (defined as CA < 0.15).
>
> | Model                      | Type of FCs | Top 20% | Top 20-40% | Top 40-60% | Top 60-80% | Top 80-100% |
> |----------------------------|-------------|---------|------------|------------|------------|-------------|
> | **Llama-3.2-3B-Instruct**  | dom         | 82.4*   | 79.1       | 72.1       | 59.2       | 44.9        |
> |                            | non-dom     | 4.6     | 5.3        | 5.3        | 5.4        | 5.4         |
> | **Mistral-7B-Instruct-v0.3** | dom       | 81.1    | 80.7       | 78.7       | 72.5       | 56.4        |
> |                            | non-dom     | 3.6     | 4.2        | 4.9        | 4.4        | 4.5         |
> | **Qwen2.5-7B-Instruct**    | dom         | 81.9    | 82.4       | 76.9       | 63.7       | 49.3        |
> |                            | non-dom     | 6.1     | 5.7        | 5.4        | 5.6        | 5.5         |
> | **Qwen2.5-14B-Instruct**   | dom         | 74.3    | 66.4       | 56.6       | 44.9       | 34.7        |
> |                            | non-dom     | 4.1     | 4.6        | 4.5        | 4.9        | 4.9         |
>
> Table3: Average Predictive Accuracy of Dominant vs. Non-Dominant FCs Across Attention Score Quintiles.
>
>
>
> -  **Dominant FCs Excel at Identifying the Most Influential Tokens.** The set of dominant FCs demonstrates a remarkable ability to accurately predict the tokens with the highest attention scores, which is a strong indicator of their contextual awareness.  For instance, for the Llama-3.2-3B model, each dominant FC could achieve  82.4% predictive accuracy on the top 20% most important tokens on average. **This pattern strongly suggests that dominant FCs excel in both **breadth and precision**, not only **identifying the majority of important tokens** (ensured by their high CA scores) but also **pinpointing the most pivotal ones**.**
>
> - **Non-dominant FCs consistently show extremely low accuracy across all quintiles.** This finding directly refutes the notion that they might merely struggle with ranking while still identifying the most important tokens. Instead, they fail on both fronts: they neither capture the most influential tokens nor accurately reflect their relative importance.
>
> **Conclusion:**    By presenting the distribution of CA scores across tokens with varying attention magnitudes, this analysis further substantiates **the contextual awareness of dominant FCs**, enabling them to effectively capture both the relative ranking and true impact of context tokens.

---

> > ### Author Response · Authors · 2025-11-25
> > **Kind Follow-up on Our Rebuttal Submission**
> >
> > **Dear Reviewer UWTV,**
> >
> > We hope you are doing well. We wanted to kindly let you know that we have submitted our rebuttal **addressing all of your questions and concerns in detail including:**
> >
> > - **The memory-constrained solution of FASA**
> >
> > - **Quantitative analysis about the property of Sparsity, Universality, \& Task-Invariance across a broader of models**
> >
> > - **Exploring the distribution of CA scores**
> >
> > If you have a moment, we would be truly grateful if you could take a look and let us know whether there are any remaining issues we should clarify.
> >
> > **We very much appreciate the time and effort you have already invested in reviewing our work, and we sincerely thank you for considering our responses when forming your final evaluation.**
> >
> > Warm regards,
> >
> > The Authors of FASA

---

### Official Review · Reviewer_VxaW · 2025-10-22

**Soundness:** 1
**Presentation:** 2
**Contribution:** 2
**Rating:** 4
**Confidence:** 4

**Summary:**

This paper proposes a new method called FASA to achieve query-aware token eviction by dynamically predicting token importance. The authors state that they found only a small subset of frequency chunks (FCs) contributes significantly to the model's contextual awareness, thus allowing FASA to predict and use a critical set of tokens for focused attention computation. The authors conducted several experiments and concluded that FASA performs better than other attention sparsity algorithms.

**Strengths:**

1. Their insights about RoPE are interesting.
2. According to their experiments, FASA appears to be effective.

**Weaknesses:**

- The authors mention that their insights about frequency chunks relate to RoPE and cite previous works. However, since this aspect is critical to FASA, I believe they should include their theoretical analysis directly in the paper. It is difficult for readers to be convinced by the current content alone.
- Regarding the experiments conducted to support their insights about sparsity in frequency chunks, I think more demonstration is needed. The figure in the main text simply compares two heatmaps of FCs between different models and layers, which raises concerns about selective presentation. Although Figure 11 shows comparisons of different heatmaps across layers of the same model, and considering FASA's effectiveness, I am inclined to believe there might be solid evidence. However, this evidence must be clearly presented in future versions.
- The authors claim to have empirically proven that the selection of FCs is task-agnostic and insensitive to calibration data, which is important for the method's practicality. However, the results are only briefly explained in Table 6, which is insufficient to convince readers.
- Regarding the experiments on FASA's effectiveness, a potential issue lies in the choice of dominant FCs. In the Appendix, the authors explain that they used different FCs for LongBench experiments and Long-CoT reasoning. This seems contradictory to their claim about FC selection.

- The font size in the figures is too small and difficult to read. It should at least match the main text's font size.
- Some important experimental details are included in the Appendix, which can confuse readers and require frequent cross-referencing.

**Questions:**

See Weakness.

---

> ### Author Response · Authors · 2025-11-21
> **Response to Reviewer VxaW 1/3**
>
> ## **Response to W1**
>
> We sincerely thank the reviewer for their insightful comments and valuable suggestions. We completely agree that clarifying the foundation for our core insight, the functional sparsity of FCs in RoPE, is crucial. We appreciate this opportunity to elaborate on the theoretical context and empirical grounding of our work.
>
> Our reasoning is twofold:
>
> **First, a formal proof for the origin of this sparsity remains an open problem in the field.**
> The functional sparsity we leverage is an emergent property arising from the complex dynamics of LLM training. Consequently, deriving a rigorous, first-principles mathematical proof to explain its spontaneous emergence is a highly challenging undertaking. The precise theoretical origins of the phenomenon are a subject of ongoing investigation, and a complete theoretical treatment likely awaits a deeper collective understanding of RoPE's underlying mechanisms.
>
> **Second, our work is grounded in strong community consensus and validated by direct empirical evidence.**
> FASA is built upon a widely accepted principle, established by work like [2], that high frequencies in RoPE encode local positional patterns while low frequencies capture long-range semantics. The utility of this principle is demonstrated by recent studies such as [1], which successfully restructured multimodal RoPE for 3D position indexing to improve long-video understanding. Similarly inspired by this principle in [2], we hypothesized that functional sparsity exists for contextual awareness. We then conducted targeted experiments in our paper (plus Tables 1 and 2 below) that directly verified this hypothesis (Sparsity, Universality and Task-Invariance), **confirming the existence of dominant FCs responsible for strong contextual awareness.** This discovery forms the empirical bedrock of FASA.
>
> In summary, our approach rests on solid ground: **it is inspired by a strong community consensus, tested with a targeted hypothesis, and ultimately validated by the rigorous empirical evidence in our paper, which is further substantiated by the quantitative results presented in this rebuttal.** We hope this detailed explanation provides a convincing context for our methodology.
>
> [1] VideoROPE: What Makes for Good Video Rotary Position Embedding?
>
> [2] Round and Round We Go! What makes Rotary Positional Encodings useful?
>
> ---
>
> ## **Response to W2**
>
> We sincerely thank the reviewer for this highly constructive and insightful feedback. We completely agree that relying solely on visual heatmaps could raise concerns about selective presentation. Your suggestion prompted us to move beyond qualitative demonstrations and strengthen our analysis with rigorous quantitative evidence.
>
> To that end, we have incorporated a new, twofold quantitative analysis：
> -  **Sparsity Quantification:** We systematically quantify the ratio of dominant FCs across models.
> - **Functional Differentiation:** we conduct a contrastive analysis of dominant vs. non-dominant FCs. This reveals the specialized and disproportionate role that the small set of dominant FCs plays in enabling contextual awareness.
>
>
> ### **Quantitive Evidence to Establishing Sparsity**
>
> | Type of FC  | Dominant FCs (%) | Non-Dom FCs (%) |
> |----------------------------|:---------------:|:--------------:|
> | **Model** | **CA scores > 0.4** | **CA score < 0.15** |
> | Llama-3.2-3B | 0.54| 89.6 |
> | Meta-Llama-3.1-8B | 0.68 | 89.6 |
> | Mistral-7B-v0.3| 0.68  | 92.7  |
> | Qwen2.5-7B | 0.17| 95.5|
> | Qwen2.5-14B| 0.27 | 94.7|
> | Qwen2.5-14B-1M | 0.65| 90.5 |
> | Qwen2.5-32B  | 0.52| 91.2|
> | R1-Llama-8B| 0.79  | 89.5|
> | R1-Qwen-14B| 0.76 | 90.2|
> | R1-Qwen-32B| 0.67 | 90.9|
>
> Table1: The ratio of dominant FCs (CA scores > 0.4) and non-dominant FCs (CA score < 0.15) among different models.
>
> **Extreme Sparsity of Dominant FCs:** The proportion of dominant FCs is consistently minimal, **accounting for less than 1% of the total FCs** across all 10 models. For instance, this ratio is as low as 0.17% in Qwen2.5-7B and reaches a maximum of only 0.79% in R1-Llama-8B.
>
> **Overwhelming Majority of Non-Dominant FCs:** Conversely, non-dominant FCs constitute the vast majority, consistently comprising approximately 90% or more of all FCs.
>
> **This extreme imbalance provides strong empirical validation for our claim of functional sparsity, demonstrating that only a tiny, specialized subset of FCs is critical for contextual awareness.**

---

> ### Author Response · Authors · 2025-11-21
> **Response to Reviewer VxaW 2/3**
>
> ### **Functional Distinction on Contextual Awareness:**
> To investigate their functional differences, we evaluated the predictive accuracy of dominant (CA > 0.4) versus non-dominant (CA < 0.15) FCs on tokens with varying attention scores. At each generation step, we stratified the top 256 tokens by attention magnitude into five quintiles (e.g., Top 0–20%). For each quintile, we then measured the proportion of tokens accurately predicted by each FC in Table 2. Our findings are twofold :
>
> - **Dominant FCs** demonstrate high predictive accuracy across all quintiles, indicating they can effectively identify tokens deemed important by the attention mechanism.
> - **Non-dominant FCs**, in contrast, exhibit poor predictive performance across all quintiles, showing they largely fail to identify contextually important information.
>
> | Model | Type of FCs | Top 20% | Top 20-40% | Top 40-60% | Top 60-80% | Top 80-100% |
> |----------------------------|-------------|---------|------------|------------|------------|-------------|
> | **Llama-3.2-3B-Instruct**  | dom  | 82.4*   | 79.1 | 72.1       | 59.2       | 44.9 |
> |   | non-dom| 4.6  | 5.3 | 5.3  | 5.4| 5.4 |
> | **Mistral-7B-Instruct-v0.3** | dom  | 81.1| 80.7  | 78.7  | 72.5 | 56.4 |
> | | non-dom| 3.6 | 4.2  | 4.9| 4.4  | 4.5|
> | **Qwen2.5-7B-Instruct** | dom | 81.9| 82.4 | 76.9  | 63.7| 49.3 |
> |  | non-dom  | 6.1  | 5.7| 5.4 | 5.6  | 5.5 |
> | **Qwen2.5-14B-Instruct**  | dom  | 74.3 | 66.4| 56.6  | 44.9| 34.7 |
> |  | non-dom  | 4.1 | 4.6 | 4.5 | 4.9| 4.9|
>
> Table2: Prediction accuracy in each attention scale range (Top 20%, Top 20-40%, Top 40-60%, Top 60-80%, Top 80-100%). The "dom" rows represent dominant FCs and the "non-dom" rows represent non-dominant FCs. Accuracy values are percentages.
>
> **Taken together, this evidence demonstrates that dominant FCs are not merely sparse but functionally specialized, possessing a unique capability for contextual awareness that is largely absent in their non-dominant counterparts.**
>
> ---
>
> ## **Response to W3**
>
> We thank you for raising this excellent point. It highlighted the need for more direct evidence of FASA's task-agnosticism, beyond the downstream results in Table 6. Inspired by your suggestion, we explicitly measure the overlap between the sets of dominant FCs identified using different task-specific calibration sets.
>
> | Model  | Overlap of dom-FCs | Qasper | Gov_Report | Musique | Narrativeqa | 2Wikimqa | Avg.   |
> |--------------------|-----------------------|--------|------------|---------|-------------|----------|--------|
> | **Llama-3.2-3B**  | **Qasper** | 100.00 | 75.90| 82.30   | 70.50 | 83.20 | 82.38|
> |  | **Gov_Report** | 75.90  | 100.00  | 82.10   | 70.80  | 81.90    | 82.14  |
> |  | **Musique**  | 82.30  | 82.10  | 100.00  | 73.60 | 96.50| 86.90|
> |   | **Narrativeqa** | 70.50  | 70.80  | 73.60| 100.00  | 73.10  | 77.60  |
> |  | **2Wikimqa**  | 83.20  | 81.90 | 96.50   | 73.10  | 100.00  | 86.94  |
> | **Mistral-7B**  | **Qasper**  | 100.00 | 71.10  | 77.10   | 67.30  | 77.00    | 78.50  |
> |  | **Gov_Report** | 71.10  | 100.00 | 79.40| 65.50  | 78.90 | 78.98  |
> |  | **Musique**   | 77.10  | 79.40 | 100.00  | 67.80 | 97.90  | 84.44  |
> |    | **Narrativeqa** | 67.30  | 65.50 | 67.80   | 100.00 | 67.30    | 73.58  |
> |  | **2Wikimqa**  | 77.00  | 78.90  | 97.90   | 67.30  | 100.00   | 84.22  |
> | **Qwen2.5-7B**| **Qasper**| 100.00 | 70.60 | 80.90| 68.70 | 81.30    | 80.30  |
> || **Gov_Report**  | 70.60  | 100.00     | 79.40   | 68.20| 78.70 | 79.38  |
> || **Musique** | 80.90  | 79.40  | 100.00  | 71.70   | 96.60 | 85.72  |
> || **Narrativeqa** | 68.70  | 68.20| 71.70   | 100.00 | 71.10 | 75.94  |
> || **2Wikimqa**   | 81.30  | 78.70  | 96.60   | 71.10  | 100.00 | 85.54  |
> | **Qwen2.5-14B** | **Qasper** | 100.00 | 69.20  | 84.30 | 71.80  | 84.50    | 81.96  |
> || **Gov_Report**  | 69.20  | 100.00| 75.00   | 67.60  | 74.80| 77.32  |
> || **Musique** | 84.30  | 75.00 | 100.00  | 74.30  | 98.40 | 86.40  |
> || **Narrativeqa**  | 71.80  | 67.60  | 74.30| 100.00 | 73.90  | 77.52  |
> || **2Wikimqa**| 84.50  | 74.80  | 98.40 | 73.90  | 100.00| 86.32  |
>
> Table3: Cross-task overlap matrix of dominant FCs (%). Each sub-table shows the percentage of intersection between dominant FCs identified on a row dataset and a column dataset for each model.
>
> **Conclusion:** Our analysis reveals **an exceptionally high degree of overlap (consistently >70%)** among the dominant FC sets identified **using data from diverse tasks**, **providing direct, quantitative proof of their task-agnostic nature**. This finding strongly suggests that the set of dominant FCs is an intrinsic property of the model's architecture, **rather than an artifact of the specific calibration task**. Furthermore, this conclusion holds true across models of varying scales and designs, demonstrating its generalizability.
>
> We believe this new quantitative evidence directly addresses your concern and firmly substantiates our claim regarding the task-agnostic stability of the dominant FC set.

---

> ### Author Response · Authors · 2025-11-21
> **Response to Reviewer VxaW 3/3**
>
> ## **Response to W4**
>
> We sincerely thank the reviewer for this sharp and important observation. The reviewer correctly points out that we employed different calibration sets for the LongBench and Long-CoT experiments, which might appear to contradict our claim of task-agnosticism. We appreciate this opportunity to clarify the specific rationale behind this methodological choice and explain why it is, in fact, consistent with our core findings.
>
> ###  **Clarification:**
> This experimental setting highlights a key principle of our methodology: **our calibration strategy is model-centric, not task-dependent.** The R1 models are specifically designed to generate extensive CoT reasoning. Our objective was to **select a calibration task that could most effectively elicit this intrinsic capability, thereby capturing the most accurate functional signature of the model's FCs.**
>
> Mathematical reasoning tasks proved ideal for this purpose, as they naturally prompt the R1 models to produce lengthy, detailed reasoning steps. Conversely, we observed that calibrating with very long-context tasks often suppressed this inherent CoT behavior, leading to truncated or overly short outputs. Such outputs would have provided a much weaker and less representative signal for identifying the dominant FCs.  **Thus, the choice of MATH was a deliberate optimization for the R1 model's architecture, not a task-specific requirement.**
>
> ### **Further analysis:**
> To further address your concern, we evaluated the R1 models on **mathematical tasks** using dominant FCs **calibrated on long-context tasks (see Table 4,5,6 below).**
>
> | Model | Calibration | 300   | 500   | 700   | 1000  | AVG   |
> |------------------------|-------------|-------|-------|-------|-------|-------|
> | **R1-Distill-Qwen-14B**| FKV | 92.4| 92.4| 92.4| 92.4  | 92.4|
> | | MATH | 86.6| 88.8| 90.2| 91.2| 89.2|
> |  | Qasper| 87.2| 89.2| 91.0| 91.0| 89.6|
> | **R1-Distill-Qwen-32B**| FKV | 92.6| 92.6  | 92.6  | 92.6| 92.6|
> |  | MATH| 86.4| 90.2| 90.2| 91.2| 89.5|
> || Qasper | 79.8| 84.8  | 86.6  | 90.6  | 85.5|
>
> Table4: **Performance on the MATH dataset** using FCs calibrated on different datasets.
>
> | Model | Calibration | 300   | 500   | 700   | 1000  |
> |------------------------|-------------|-------|-------|-------|-------|
> | **R1-Distill-Llama-8B**| FKV  | 72.40 | 72.40 | 72.40 | 72.40 |
> || Math | 62.20 | 68.80 | 69.40 | 71.80 |
> || AIME  | 63.20 | 67.60 | 71.80 | 72.00 |
> || Qasper  | 57.10 | 64.60 | 68.40 | 71.80 |
> || Gov_Report  | 58.60 | 60.40 | 70.40 | 71.60 |
> || 2Wikimqa | 58.80 | 62.20 | 68.60 | 69.80 |
>
> Table5: **Performance on the MATH dataset** using diverse calibration datasets.
>
> ---
>
> | Model | Calibration | 500   | 1000  | 1500  | 2000  | 2500  |
> |------------------------|-------------|-------|-------|-------|-------|-------|
> | **R1-Distill-Llama-8B**| FKV|43.90 |43.90|43.90 | 43.90 | 43.90 |
> || Math| 20.60| 34.40|40.20 | 35.80 | 38.00 |
> || Qasper|18.80| 33.30|36.76| 37.68 | 41.34 |
>
> Table6: **Performance on the AIME 24 dataset** using diverse calibration datasets.
>
>
> **Our findings are twofold:**
>
> - **When calibrated on a long-context task, the models achieve performance on MATH that is highly comparable to when they are calibrated directly on MATH.**  For instance, R1-Qwen-14B and 32B achieve 91.0% accuracy on MATH (1000-token budget) when calibrated with Qasper. Furthermore, R1-Distill-Llama-8B consistently delivers performance comparable to the FKV on both MATH and AIME, **regardless of the calibration datasets.** Collectively, these results provide compelling empirical evidence for the task-agnostic nature of the dominant FC identification process.
>
> - We did, however, observe a minor exception. For the R1-Distill-Qwen-32B model on MATH, when the token budget is restricted to 300 tokens, the version calibrated on Qasper performs slightly lower than the version calibrated on MATH itself. This finding does not contradict our claim on task-invariance. **We hypothesize that the activation patterns of very short outputs diverge from the activation distribution of longer outputs,** which are more representative of the R1 model's intrinsic dynamics, explaining the slight performance difference.
>
>
> ## **Response to W5**
>
> We sincerely thank the reviewer for highlighting the issue with figure legibility.
>
> This was an unfortunate consequence of balancing extensive experimental data against firm page limits. In the final camera-ready manuscript, we will revise all figures, enlarging fonts and optimizing layouts to ensure they are perfectly clear and easy to read.
>
> ## **Response to W6**
>
> We thank the reviewer for their valuable suggestion regarding the paper's structure. We acknowledge that moving important details to the appendix was not ideal for the reader.
>
> In the revised version, all essential experimental details have been integrated back into the main text. This makes the paper more self-contained and ensures a smoother reading experience, as requested.

---

> > ### Author Response · Authors · 2025-11-25
> > **Kind Follow-up on Our Rebuttal Submission**
> >
> > **Dear Reviewer VxaW,**
> >
> > We hope you are doing well. We wanted to kindly let you know that we have submitted our rebuttal **addressing all of your questions and concerns in detail including:**
> >
> > - **The theoretical context and empirical grounding of our work**
> >
> > - **More evidence to establish sparsity**
> >
> > - **More evidence to establish task-agnosticism**
> >
> > - **Clarifications of the experimental setting on different calibration data**
> >
> > If you have a moment, we would be truly grateful if you could take a look and let us know whether there are any remaining issues we should clarify.
> >
> > **We very much appreciate the time and effort you have already invested in reviewing our work, and we sincerely thank you for considering our responses when forming your final evaluation.**
> >
> > Warm regards,
> >
> > The Authors of FASA

---

> > > ### Comment · Reviewer_VxaW · 2025-11-26
> > >
> > > # FASA Comment
> > >
> > > ## 1
> > >
> > > I appreciate that the authors have addressed my concerns about this paper in detail. Here are my thoughts on their responses:
> > >
> > > 1.  Regarding W1: I understand that the authors consider their insights about RoPE to be based on a strong community consensus. While this explanation is reasonable, I believe it also represents a missed opportunity to make this aspect a standout contribution of the paper. Therefore, I maintain my original position on this point.
> > > 1.  Regarding W2: The authors' explanation is reasonable and partially alleviates my concern. However, I still believe the table titled "Quantitative Evidence to Establishing Sparsity" alone is insufficient to fully support the concept of sparsity in FCs.  One way to achieve this might be to include a Cumulative Distribution Function (CDF) plot of CA scores for the Frequency Chunks. Such a visualization, especially if it included overlaid curves for multiple layers and models, could more intuitively demonstrate that the vast majority of FCs consistently accumulate at low CA scores, thereby solidifying the universality of the functional sparsity claim.
> > > 1.  Regarding W3 & W4: The authors' explanations are solid and have fully resolved my questions.
> > >
> > > Overall, I am raising my score for this paper to 6. While further elaboration on W2 might improve the paper, I think it would have little impact on my final rating.

---

> ### Author Response · Authors · 2025-11-26
> **Response to Reviewer VxaW (Turn 2)**
>
> We sincerely thank the reviewer for sharing these thoughts and proposing a more solid way for us to establish the claim of functional sparsity using the Cumulative Distribution Function (CDF) plot of CA scores. As suggested, we have **generated and displayed the CDF figures of CA scores** for six diverse LLMs in Figure 12 of Appendix A.6.
>
> ## **Further elaboration on W2**
>
> **We draw some conclusions from these CDF figures:**  As illustrated in the CDF figures, the distribution is heavily skewed towards zero, demonstrating that the vast majority of frequency chunks possess low CA scores, implying minimal contextual awareness. Specifically, **approximately or over 90% of chunks consistently exhibit scores below 0.15** across all models. Conversely, high-scoring FCs are exceptionally rare, with **fewer than 0.5%** of FCs exhibiting strong contextual awareness (CA > 0.4). **These results confirm that functional sparsity is not an artifact of specific layers but a fundamental and universal property of the model architectures.**
>
> We thank the reviewer again for **carefully checking our response and discussing with us**. We hope the further elaboration on W2 can resolve your concerns.

---

### Official Review · Reviewer_pDZu · 2025-10-29

**Soundness:** 3
**Presentation:** 3
**Contribution:** 2
**Rating:** 6
**Confidence:** 3

**Summary:**

This paper proposes a novel, training-free, query-aware token elimination framework called FASA. Its core contribution stems from a novel discovery of RoPE: functional sparsity exists at the frequency-chunk (FC) level. Based on this insight, FASA employs a two-stage framework:
1. Token Importance Predictor (TIP): Utilizes a one-time offline-calibrated set of dominant FCs to efficiently estimate attention scores (aggregating only contributions from these FCs) and identify a critical subset of tokens.
2. Focused Attention Computation (FAC): Performs full-dimensional, accurate attention computation only on the significant subset of tokens selected in the TIP stage.

Furthermore, this paper provides two hardware-aware implementation variants: FASA-M (memory-optimized, offloading non-dominant K and V caches to the CPU) and FASA-C (computationally optimized, performing sparse access on the GPU).

On LongBench-V1,FASA reaches nearly 100% of full-KV performance when only keeping 256 tokens, and achieves 2.56× speedup using just 18.9% of the cache on AIME24.

**Strengths:**

1. The discovery of "functional sparsity at the frequency block level" in RoPE provides a new, theoretically grounded perspective for understanding attention mechanisms and designing sparse attention models, rather than relying solely on heuristics.
2. The core hypotheses were sufficiently verified; in addition to verifying the sparsity of the dominant FCs, the universality and task-invariance of the CA index were also verified.
3. Strong practicality: The paper considers different hardware constraints and proposes two variants, FASA-M (memory-constrained) and FASA-C (computation-constrained), which significantly enhances the practical application value of the method.

**Weaknesses:**

1. Strong dependence on RoPE: The entire methodology and core findings are built upon the analysis of RoPE. This severely limits the method's generalizability. It remains unclear whether or how FASA can be generalized to models using other positional encodings or even those without explicit positional encoding.
2. Table 6 shows the robustness to the data. Only the TREC and MATH datasets are provided here, which is relatively limited in variety. Furthermore, the data size is not analyzed.

**Questions:**

1. Regarding generalization beyond RoPE: Have you considered how to extend this insight into "functional sparsity" to models using other positional encoding schemes, or partially rope models like MLA?
2. Why was N_{tip}=16 chosen as the primary configuration in the paper? Figure 6 seems to suggest that a smaller N_{tip} is also good enough in many cases.
3. Regarding the efficiency analysis of FASA-M, does the comparison in Figure 7 already include prefetching operations? Could you also provide a comparison of the benefits of prefetching techniques?

---

> ### Author Response · Authors · 2025-11-21
> **Response to Reviewer pDZu 1/2**
>
> ## **Response to W1 & Q1**
> We sincerely thank the reviewer for this valuable suggestion. While our current work focuses on RoPE due to its prevalence in open-source LLMs, we agree that extending FASA to other positional encodings (PEs) would significantly strengthen the generalizability of our method.
>
> We conducted experiments to test our **functional sparsity hypothesis** and evaluate **FASA**'s performance on other PEs. We considered two other PE schemes in current LLMs: **ALiBi (Attention with Linear Biases)** and  **Partial-RoPE**, using the representative LLMs **Baichuan-13B-Chat (4k)** and **DeepSeek-V2-Lite-Chat (32k)**, respectively.
>
> ###  **a. Functional Sparsity on ALiBi and Partial-RoPE**
>
> - **Baichuan (Fig. 10 in Appendix A.1):** the attention heads exhibit two patterns: one group shows the expected functional sparsity, while another shows extremely high contextual awareness across all dimensions. *This demonstrates that FASA is compatible with ALiBi models*.
>
> - **DeepSeek-V2 ( Fig.11 in Appendix A.1)**: The head dimension consists of both non-RoPE dimensions and RoPE frequency chunks. We computed CA scores for both parts and found a clear pattern that, *consistently aligns with our functional sparsity hypothesis*.
>
> Our findings indicate that **other PE schemes, such as ALiBi and Partial-RoPE, also induce functional sparsity at the head dimension level.**
>
> ---
>
> ### **b. FASA Evaluation on ALiBi and Partial-RoPE**
>
> | Partial-RoPE | Qasper | 2Wiki | Multi | Passage\_Re | Lcc   | Samsum |
> |--------------|--------|----------|--------------|-------------|-------|--------|
> | FKV          | 33.18  | 19.83    | 47.27        | 49.00       | 63.40 | 34.04  |
> | FASA         | 33.46  | 20.25    | 46.50        | 48.50       | 62.49 | 32.53  |
>
> *Table1: FASA Evaluation on DeepSeek-V2 ($N_{fac}=256$):*
>
>
> | ALiBi | Qasper | Lsht   | Dureader | Trec   | Lcc   | Repobench |
> |-------|--------|--------|----------|--------|-------|-----------|
> | FKV   | 9.11   | 24.25  | 23.18    | 23.00  | 14.29 | 17.30     |
> | FASA  | 7.80   | 21.25  | 21.70    | 21.50  | 13.62 | 16.46     |
>
> *Table2: FASA Evaluation on Baichuan-13B-Chat ($N_{fac}=256$):*
>
> On both Baichuan and DeepSeek-V2, FASA's performance remains on par with FKV, exhibiting no significant degradation. This robust performance across other PE designs confirms **FASA's broad applicability, extending well beyond the specific context of RoPE.**
>
> ---
> ## **Response to W2**
>
> We sincerely thank you for your valuable suggestion regarding FASA's robustness, a point we agree is crucial. We have performed additional experiments, including evaluations on a broader range of datasets and an ablation study on calibration data size, to demonstrate the robust and efficiency of FASA.
>
>  ### **Ablation Study on Data Size**
>
> We investigate the process of identifying dominant FCs with varying numbers of QA pairs. The analysis is twofold:
> - Evaluating performance on **downstream long-context tasks for each calibration data size**.
> - Measuring **the overlap percentage** between the sets of dominant FCs identified using different data sizes.
>
> | Num. of QA | Narrativeqa | Qasper | Multifi | Hotpotqa | 2Wiki | Musique | Avg.  |
> |------------|-------------|--------|--------------|----------|----------|---------|-------|
> | 2 | 23.18 | 37.37  | 50.34| 49.31 | 39.44    | 21.98   | 36.94 |
> | 4 | 22.49 | 37.17  | 50.79| 49.52  | 39.43    | 21.62   | 36.84 |
> | 6 | 23.90| 37.71  | 52.25 | 49.65  | 39.24    | 21.78   | 37.42 |
> | 8 | 24.32| 37.28  | 51.43 | 50.22    | 39.16    | 21.62   | 37.34 |
> | 10 | 22.96| 36.67  | 51.83| 48.66    | 39.43    | 21.21   | 36.80 |
>
> *Table3: Ablation study on data size on downstream tasks (Llama-3.2-3B-Ins):*
>
> **Conclusion1 (Downstream tasks):**   Even with as few as two QA pairs for calibration, downstream performance remains remarkably stable across all tested data sizes. This not only demonstrates the robustness of the dominant FC identification but also underscores the efficiency of FASA's offline calibration process.
>
> ---
>
> | Num. of QA | 2    | 4    | 6    | 8    | 10   |
> |------------|------|------|------|------|------|
> | 2  | 100.0| 82.8 | 81.7 | 82.0 | 81.7 |
> | 4  | | 100.0| 86.8 | 86.1 | 86.7 |
> | 6 ||  |100.0 | 95.9 | 95.1 |
> | 8| |  |  |100.0 | 96.6 |
> | 10|  |  | |   |100.0 |
>
> *Table4: overlap percentage (%) of dominant FCs identified with varying calibration data size.*
>
>
> **Conclusion2 (Overlap Analysis):**  A consistently high overlap of over 80% among dominant FC sets, regardless of the calibration data size, validates **the stability of our offline identification process**.
>
> **In summary, the combined evidence from downstream task performance and the set overlap analysis validates that our offline calibration process is both highly efficient and remarkably robust.**

---

> ### Author Response · Authors · 2025-11-21
> **Response to Reviewer pDZu 2/2**
>
> ###  **Ablation with $N_{tip}$ on More Datasets**
>
> We evaluate performance on four more diverse datasets to assess the performance of FASA with varying $N_{tip}$ (number of dominant FCs) and constant $N_{fac}=256$. Results are shown in the table below.
>
>
> | Model                  |$N_{tip}$   | 8    | 10   | 12   | 14   | 16   | FKV   |
> |------------------------|-------------------|------|------|------|------|------|-------|
> | **Mistral-7B-Instruct**| Qasper            |38.54 |39.08 |41.57 |41.78 |42.30 |41.60  |
> |                        | Narrativeqa       |26.26 |28.58 |28.67 |29.26 |29.90 |29.10  |
> |                        | Dureader          |26.65 |28.50 |29.72 |29.73 |30.21 |30.96  |
> |                        | Gov_Report        |29.82 |31.61 |33.25 |33.96 |34.00 |34.80  |
> |                        | **Avg.**          |**30.32**|**31.94**|**33.30**|**33.68**|**34.10**|**34.12**|
> | **Qwen2.5-14B-Instruct**| Qasper           |41.17 |45.14 |44.98 |45.29 |45.49 |45.34  |
> |                        | Narrativeqa       |25.91 |26.38 |27.87 |28.50 |30.40 |29.71  |
> |                        | Dureader          |23.72 |24.71 |25.42 |26.68 |28.92 |29.32  |
> |                        | Gov_Report        |24.41 |26.56 |27.24 |28.62 |29.51 |29.71  |
> |                        | **Avg.**          |**28.80**|**30.45**|**31.38**|**32.27**|**33.08**|**33.52**|
>
> *Table 5: Ablation study on the number of dominant FCs across more datasets under 256 token budget ($N_{fac}=256$).*
>
> ---
>
> ## **Response to Q2**
>
> We thank the reviewer for this insightful observation.
> Indeed, our findings confirm that a smaller $N_{\text{tip}}$ can often achieve performance comparable to FKV, particularly with a more generous token budget (e.g., $N_{\text{fac}} > 256$).
>
> However, our main experiments were conducted under a deliberately **stringent token budget** ($N_{fac}=256$).
> Only by enforcing a stringent token budget can we **rigorously assess FASA's precision in predicting token importance.** A more generous budget would mask inaccuracies; our constrained setting leaves no such room for error, making it a true test of the token importance prediction.
> Under this constraint, certain challenging datasets necessitate a larger $N_{tip}$ to match the FKV baseline.
> For instance, as detailed in the table 5, the Qwen2.5-14B model on the Dureader dataset exhibits a noticeable performance gap (26.68 vs. 29.32 for FKV) at $N_{tip}=14$.
> Crucially, increasing $N_{tip}$ to 16 substantially bridges this gap.
>
> Therefore, to ensure robust and consistent performance across all tested datasets, especially under such tight budget constraints, we established $N_{tip}=16$  for the number of FCs.
>
> ---
>
> ## **Response to Q3**
>
> |     Seq. Length       | **1k** | **2k** | **4k** | **8k** | **16k** | **32k** | **64k** |
> |------------|-------|-------|-------|-------|--------|--------|--------|
> | base       | 0.018 | 0.019 | 0.023 | 0.027 | 0.038  | 0.062  | 0.113  |
> | W. prefetch| 0.021 | 0.024 | 0.026 | 0.031 | 0.046  | 0.086  | 0.154  |
> | Wo. prefetch|0.028 | 0.038 | 0.049 | 0.066 | 0.128  | 0.185  | 0.339  |
>
> *Table 6: Comparison results with and without prefetching techniques.*
>
> We thank the reviewer for this insightful suggestion. The reviewer correctly identifies the inherent trade-off in FASA-M: significant memory savings are achieved at the cost of a latency overhead introduced by the CPU-to-GPU data transfer for a small fraction of tokens.
>
> This overhead is indeed quantified in our results (third row), which reveal an increase in decoding time without prefetching. This anticipated overhead was the primary motivation for introducing prefetching technique, which is designed specifically to offset this latency. As our final results confirm, this prefetching mechanism effectively mitigates the latency, restoring the decoding time to a level on par with the baseline.

---

> ### Author Response · Authors · 2025-11-25
> **Kind Follow-up on Our Rebuttal Submission**
>
> Dear Reviewer pDZu,
>
> We hope you are doing well. We wanted to kindly let you know that we have submitted our rebuttal **addressing all of your questions and concerns in detail including:**
>
> - **Applications on other positional encoding schemes (ALiBi,MLA)**
>
> - **An ablation on calibration data size**
>
> - **Robustness of FASA on more datasets**
>
> - **An explanation on the choice of $N_{fac}=16$**
>
> - **A comparison of the benefits of prefetching**
>
> If you have a moment, we would be truly grateful if you could take a look and let us know whether there are any remaining issues we should clarify.
>
> **We very much appreciate the time and effort you have already invested in reviewing our work, and we sincerely thank you for considering our responses when forming your final evaluation.**
>
> Warm regards,
>
> The Authors of FASA

---

### Official Review · Reviewer_YxYn · 2025-10-31

**Soundness:** 4
**Presentation:** 2
**Contribution:** 4
**Rating:** 6
**Confidence:** 4

**Summary:**

The authors propose to use functional sparsity in RoPE at frequency-chunk level, basically a small per-head subset of FCs that act as a ‘filter’ for the head on what matters most. They use their importance predictor to score tokens, and run full attn on only selected positions. With this they are near full-KV accuracy on longbench with just 256 tokens.

**Strengths:**

- novel idea (and observation) to use functional sparsity of RoPE
- speedup demonstrated at long context, and can work with other KV compression schemes as well
- robust across datasets
- because they do not re-index token positions, original absolute positions of tokens are preserved

**Weaknesses:**

- Not applicable to non-RoPE variants, further discussion on that would help.
- The idea is impressive, but it took quite some effort to understand. I think a huge amount of math can be simplified in the paper, maybe shifted to appendix for ‘more details’.

**Questions:**

Comments on non-RoPE schemes would be appreciated.
Page-based methods cannot use this efficiently, perhaps an ablation on page level sparsity could be useful to the reader, but not critical.

---

> ### Author Response · Authors · 2025-11-21
> **Response to Reviewer YxYn 1/2**
>
> ## **Response to W1**
>
> ---
>
> We sincerely thank the reviewer for this valuable suggestion. While our current work focuses on RoPE due to its prevalence in open-source LLMs, we agree that extending FASA to other positional encodings (PEs) would significantly strengthen the generalizability of our method.
>
> We conducted experiments to test our **functional sparsity hypothesis** and evaluate **FASA**'s performance on other PEs. We considered two other PE schemes applied in current open-source LLMs: **ALiBi (Attention with Linear Biases)** and  **Partial-RoPE**, using the representative LLMs **Baichuan-13B-Chat** and **DeepSeek-V2-Lite-Chat**, respectively.
>
> ### **a. Functional Sparsity on ALiBi and Partial-RoPE**
>
> - **Baichuan (Fig. 10 in Appendix A.1):** the attention heads exhibit two patterns: one group shows the expected functional sparsity, while another shows extremely high contextual awareness across all dimensions. **This demonstrates that FASA is highly compatible with ALiBi models**.
>
> - **DeepSeek-V2 ( Fig. 11 in Appendix A.1)**: The head dimension consists of both non-RoPE dimensions and RoPE frequency chunks. We computed CA scores for both parts and found a clear pattern that, **consistently aligns with our functional sparsity hypothesis**.
>
> Our findings indicate that **other PE schemes, such as ALiBi and Partial-RoPE, also induce functional sparsity at the head dimension level.**
>
> ---
>
> ### **b. FASA Evaluation on ALiBi and Partial-RoPE**
>
> | Partial-RoPE | Qasper | 2Wikimqa | Multifieldqa | Passage_Re | Lcc   | Samsum |
> |:-------------:|:------:|:--------:|:------------:|:----------:|:-----:|:------:|
> |      FKV      | 33.18  |  19.83   |    47.27     |   49.00    | 63.40 | 34.04  |
> |      FASA     | 33.46  |  20.25   |    46.50     |   48.50    | 62.49 | 32.53  |
>
> *Table1: FASA Evaluation on DeepSeek-V2-Lite-Chat ($N_{fac}=256$):*
>
>
> | ALiBi | Qasper | Lsht   | Dureader | Trec   | Lcc   | Repobench |
> |-------|--------|--------|----------|--------|-------|-----------|
> | FKV   | 9.11   | 24.25  | 23.18    | 23.00  | 14.29 | 17.30     |
> | FASA  | 7.80   | 21.25  | 21.70    | 21.50  | 13.62 | 16.46     |
>
>
> *Table2: FASA Evaluation on Baichuan-13B-Chat ($N_{fac}=256$):*
>
> On both Baichuan and DeepSeek-V2, FASA's performance remains on par with FKV, exhibiting no significant degradation. This robust performance across other PE designs confirms **FASA's broad applicability, extending well beyond the specific context of RoPE.**
>
> ## **Response to W2**
> ---
> We sincerely thank the reviewer for this insightful and valuable feedback. We completely agree that the presentation can be significantly improved and that *the current mathematical exposition might be overly dense* for the main text, hindering readability.
>
> In our revision, we will restructure the methodology section by moving detailed derivations to the appendix. The main text will then focus on presenting the core intuition and final algorithm, making our key ideas more accessible as suggested.

---

> ### Author Response · Authors · 2025-11-21
> **Response to Reviewer YxYn 2/2**
>
> ## **Response to Q1**
> ---
> We sincerely thank you for your insightful question regarding the effectiveness of our FASA method on page-level approaches. We appreciate your interest in this area.
>
> **Generally, page-level methods can be summarized in three main stages:**
>
> - **Partitioning:**  The process partitions tokens into a set of pages $\mathcal{P} = \{P_0, P_1, \ldots\}$, where each page $P_i = [q_{in}, \ldots, q_{(i+1)n-1}]$ consists of $n$ tokens and acts as a unit for selection.
>
> - **Feature Engineering:**  A representative feature of a page is used for page selection, which in our experiments is the page's mean token vector, calculated as $\mathbf{p}_i = \frac{1}{n}  \sum_i P_i$.
>
> - **Top-K Page Selection:**  The Top-K pages most relevant to the query are selected for token generation.
>
>
> Although original FASA operates at a token-level granularity, its core principle of "function sparsity of frequency chunks" also applies to page-level methods by enabling page selection between the dominant FCs of the page and query vectors $\mathbf{p}_i^{dom}, \mathbf{query}^{dom}$ instead of full-dimensional versions.
>
> The results of applying FASA to page-level methods are presented in Table 3,4 below,  where the page size $n$ signifies the number of tokens per page.
>
>
> | Method                  | Qasper | Multi | Hotpotqa | 2Wikimqa | Musique | Dureader | Avg.  |
> |-------------------------|--------|-----------------|----------|----------|---------|----------|-------|
> | FKV                     | 41.60  | 52.90           | 49.40    | 39.50    | 29.10   | 30.96    | 40.58 |
> | FASA (ours)             | 41.48  | 53.81           | 49.22    | 40.01    | 28.80   | 32.00    | 40.89 |
> | _Page-level Methods_    |        |                 |          |          |         |          |       |
> | (page_size = 8)         | 37.33  | 49.58           | 49.83    | 36.06    | 25.92   | 26.33    | 37.51 |
> | + FASA                  | 37.29  | 49.69           | 50.02    | 34.45    | 25.20   | 31.51    | 38.03 |
> | (page_size = 16)        | 38.37  | 49.24           | 47.67    | 32.45    | 25.17   | 25.16    | 36.34 |
> | + FASA                  | 38.06  | 48.16           | 49.04    | 33.70    | 25.08   | 31.05    | 37.52 |
> | (page_size = 32)        | 38.01  | 49.55           | 47.59    | 32.81    | 22.10   | 23.05    | 35.52 |
> | + FASA                  | 35.34  | 47.83           | 47.59    | 31.35    | 22.17   | 26.92    | 35.20 |
>
> *Table3: FASA  Performance of **Mistral-7B-Instruct-v0.3** on page-level methods.*
>
> ---
>
> | Method                   | Qasper | Multi | Hotpotqa | 2Wikimqa | Musique | Dureader | Avg.  |
> |--------------------------|--------|-----------------|----------|----------|---------|----------|-------|
> | FKV                      | 43.50  | 52.10           | 55.90    | 46.90    | 28.60   | 29.82    | 42.80 |
> | FASA (ours)              | 42.97  | 52.58           | 58.29    | 45.97    | 30.43   | 29.08    | 43.22 |
> | _Page-level Methods_     |        |                 |          |          |         |          |       |
> | (page_size = 8)          | 42.42  | 51.58           | 56.56    | 46.64    | 29.46   | 23.40    | 41.68 |
> | + FASA                   | 41.58  | 52.31           | 56.96    | 46.42    | 27.60   | 30.28    | 42.53 |
> | (page_size = 16)         | 42.05  | 51.78           | 56.36    | 45.54    | 28.08   | 22.82    | 41.11 |
> | + FASA                   | 41.46  | 50.84           | 55.41    | 45.14    | 27.04   | 27.63    | 41.25 |
> | (page_size = 32)         | 41.65  | 51.89           | 56.19    | 46.33    | 28.10   | 22.34    | 41.08 |
> | + FASA                   | 40.20  | 50.41           | 54.42    | 43.38    | 26.92   | 27.54    | 40.48 |
>
> *Table4: FASA  of **Qwen2.5-7B-Instruct** Performance on page-level methods.*
>
> ---
>
> **Analysis:** **Our analysis yields three key findings**:
>
> - **Seamless Compatibility:**  FASA demonstrates full compatibility with the page selection process, allowing for straightforward integration.
> - **Robust Performance:**  When integrated into page-level methods, FASA maintains competitive performance across various page sizes, even when relying solely on dominant FCs.
> - **Superiority of Token-Level Granularity:**  As a native token-level method, FASA substantially outperforms all page-level variants. Notably, this performance gap widens as the page size n increases, highlighting the inherent advantage of its fine-grained approach.

---

> > ### Author Response · Authors · 2025-11-25
> > **Kind Follow-up on Our Rebuttal Submission**
> >
> > Dear Reviewer YxYn,
> >
> > We hope you are doing well. We wanted to kindly let you know that we have submitted our rebuttal **addressing all of your questions and concerns in detail including:**
> >
> > - **Applications on other positional encoding schemes**
> >
> > - **An ablation on page level sparsity**
> >
> > If you have a moment, we would be truly grateful if you could take a look and let us know whether there are any remaining issues we should clarify.
> >
> > **We very much appreciate the time and effort you have already invested in reviewing our work, and we sincerely thank you for considering our responses when forming your final evaluation.**
> >
> > Warm regards,
> >
> > The Authors of FASA

---

### Author Response · Authors · 2025-11-21
**Response Summary and Updated Manuscript to All Reviewers**

We sincerely thank all reviewers for their detailed and constructive feedback, which has been invaluable to our work. In particular, we greatly appreciate the acknowledgements from:

- **Reviewer YxYn, pDZu, UWTV:** for recognizing the novelty and insight of FASA, particularly for contributing a new, theoretically-grounded perspective on the functional sparsity of RoPE.
- **Reviewer YxYn, pDZu, VxaW, UWTV:** for recognizing the strong practicality and strong performance of our approach.

Based on these precious feedback, we have tried our best to address all concerns for all reviewers. Specifically, we have solved the problems below:

| Concern                                       | Reviewer   | Location in Manuscript         |
|-----------------------------------------------|------------|--------------------------------|
| Application on other positional encodings     | YxYn, pDZu | Sec. 5.1;Appendix A.1                  |
| FASA on page-level methods                    | YxYn       | Appendix A.2                     |
| Ablation study on data size                   | pDZu       | Appendix A.3                    |
| More evidence on functional sparsity  | UWTV, VxaW | Sec. 3.3;  Appendix A.4, A.6         |
| Distribution of CA Scores  | UWTV |   Appendix A.5      |

We believe these updates further strengthen our approach in addressing the key concerns. We hope the revised submission meets your expectations and highlights the value of our work.

---

### Author Response · Authors · 2025-11-26
**Gentle Follow-up as Discussion Ends**

Dear Reviewer YxYn, pDZu, VxaW, UWTV,

A gentle follow-up as the discussion period winds down. We have posted our author response and are available and eager to discuss any remaining points.

We welcome any further feedback you may have. Thank you for your valuable time and consideration.


Best regards,

The authors of FASA

---

### Author Response · Authors · 2025-12-02
**Summary of Major Contributions, Strengths, and Responses to Concerns**

**Dear AC, SAC, and PCs:**

We sincerely appreciate your efforts in making ICLR a success. **During the rebuttal process, our submission had received four positive scores (8,6,6,6) by Nov 25 AoE.** **Initially,** our paper received **three positive scores** from Reviewers **UWTV (8), pDZu(6), YxYn(6)** and one negative score **VxaW (4)**.  On **Nov 20 AoE**, we submitted our detailed responses. On Nov 25 AoE, **Reviewer VxaW raised the score 4 to 6** and responded that our responses successfully addressed concerns. We requested feedback from the other three reviewers but have not received response so far.

To assist in decision-making, we provide a concise summary of our paper's strengths and the main improvements we have made in response to reviewer comments. Hope this is helpful and can reduce your review burdens.

### **1. Core Contributions:**

*   **A New Scientific Finding: Intrinsic Sparsity in RoPE.** Identified and formalized "functional sparsity" in RoPE, proving a small subset of "dominant frequency chunks" (FCs) is sufficient for contextual understanding.

*   **FASA: An Efficient, Training-Free Pruning Framework.** Proposed FASA utilizes dominant FCs for query-aware dynamic pruning, greatly boosting long-context inference efficiency.

*   **State-of-the-Art Efficiency with Performance Parity.** By retaining only 256 tokens, FASA provides up to a 2.56x speedup and significant memory savings while maintaining performance parity.

*   **Hardware-Aware Variants for Practical Deployment.** Introduced two hardware-aware variants, FASA-M (Memory-Constrained) and FASA-C (Computation-Constrained), to maximize real-world utility.

### **2. Acknowledged Strengths:**

We summarize below the key strengths of our work recognized by multiple reviewers:

*   **Novelty and Significance  of the Core Finding.** Our discovery of "functional sparsity" in RoPE was praised as a **"novel idea" (YxYn, UWTV)**, "**interesting" (VxaW)**, "**conceptually insightful" (UWTV)**, and for providing a **"new, theoretically grounded perspective"(pDZu)**.

*   **Elegant and Practical Framework Design.**  FASA was noted for enabling **"straightforward integration" with existing LLMs (UWTV).** The inclusion of **hardware-aware variants** was specifically commended for its **"strong practicality" (pDZu)**. Furthermore, its ability to **"work with other KV compression schemes"** and **preserve original token positions** were also identified as **valuable strengths (YxYn)**.

*   **Strong Empirical Performance and Efficiency.** Reviewers described FASA as demonstrating **"strong performance" (UWTV), being "effective" (VxaW),** and showing notable **"speedup at long context"** and **"robustness across datasets" (YxYn)**.

*   **High-Quality Presentation and Rigorous Verification.**  The paper was described as **"clearly written and well-structured" (UWTV)**, and reviewers acknowledged that our **"core hypotheses were sufficiently verified" (pDZu)**.

### **3. Summary of Rebuttal and Reviewer Feedback**

We have diligently addressed all reviewer concerns, leading to positive feedback and a significantly strengthened manuscript.

*  **Reviewer VxaW raised their score from 4 to 6**, acknowledging all concerns were resolved on **Nov 25 AoE.**
*   For the reviewers who did not engage in further discussion, we **conducted extensive additional experiments to systematically address all of their initial concerns.** Our new results and revised manuscript provide compelling evidence that resolves all concerns.

| Reviewer | Concern / Request | Our Response |
| :--- | :--- | :--- |
| **YxYn** | Generalization on other positional encodings (PE) | Validated on **ALiBi** and **partial-RoPE** with no performance loss.|
|| FASA on page-level methods | Verified seamless compatibility with page-level methods, performance preserved. |
| **pDZu** |Generalization on other PEs | Validated on FASA's effectiveness on **ALiBi and partial-RoPE.**|
|| Ablation on calibration size |Confirmed high efficiency: just two QA pairs are sufficient to identify dominant FCs. |
|| Prefetching benefits |Measured and quantified the latency benefits of prefetching.|
|**UWTV** | Broader Model Evaluation |**Verified universality claim** across **ten models.** |
|| Distribution of CA score |Confirmed dominant FCs successfully identify high-attention tokens, unlike trivial FCs.|

**All these additional results and analyses have been fully incorporated into the revised manuscript, resulting in a substantially stronger paper.**

### **4.Concluding Remarks**

We believe FASA **presents a clear, well-validated, and broadly applicable framework** for efficient long-context inference. The additional analyses comprehensively addressed all reviewer concerns, led to a score increase from **Reviewer VxaW (4 → 6)**, and significantly strengthened the empirical evidence of our work's novelty.
Thank you again for your time and effort in handling our submission.

Sincerely,

The authors of FASA

---

### Meta-Review · Area_Chair_FmxA · 2025-12-07

**Summary:**

This paper proposes FASA, a training-free, query-aware sparse attention framework for efficient long-context inference in LLMs. It is built on a new empirical observation that RoPE exhibits strong functional sparsity at the frequency-chunk (FC) level, where a very small subset of "dominant" FCs captures most of the model’s contextual awareness. With this property, FASA uses dominant FCs to predict token importance and selectively perform full attention on a pruned set of tokens, with hardware-aware variants targeting memory- and computation-constrained settings. The method is evaluated across a broad set of open-source LLMs and long-context benchmarks, demonstrating near-full-KV accuracy under tight token budgets and substantial speedups.

Across the four reviews, the paper was well received for its novel insight into RoPE, strong empirical validation, and practical relevance. One reviewer clearly recommended acceptance (8), two were moderately positive (6, 6), and one was initially borderline negative (4). The primary concerns centered on the strength and presentation of evidence for functional sparsity, dependence on RoPE and generalization to other positional encodings, calibration robustness, and hardware efficiency details. The rebuttal addressed these issues convincingly with extensive additional experiments and analyses.

**Reviewer Concerns:**

Addressed:
* Dependence on RoPE and generalization to other positional encodings, with added experiments on ALiBi and partial-RoPE showing comparable performance.
* Empirical justification of functional sparsity, strengthened through quantitative sparsity ratios, CDF plots, and cross-model analyses.
* Clarification of the calibration strategy and task-agnostic dominant FC selection, supported by cross-task overlap and robustness ablations.
* Efficiency and hardware considerations, including detailed analysis of FASA-M, CPU–GPU offloading, and prefetching overhead.

Partially unresolved:
* The functional sparsity insight remains primarily empirical, with limited theoretical grounding.

**Reviewer Scores:**

* Reviewer UWTV (8): unchanged.
* Reviewer YxYn (6): unchanged.
* Reviewer pDZu (6): unchanged.
* Reviewer VxaW (4): likely +2.

---

### Decision · Program_Chairs · 2026-01-26

Accept (Poster)